# Maintenance of intestinal CX3CR1+ macrophage homeostasis defines post-treatment control in SIV-infected macaques

Stéphane Hua[1,5], Keltouma Benmeziane[1], Delphine Desjardins [1], Nastasia Dimant [1], Marco Leonec[1], Laetitia Bossevot [1], Julien Lemaitre [1], Adeline Mélard [2,3], Francis Relouzat [1], Véronique Avettand-Fenoël [2,3], Nathalie Dereuddre-Bosquet [1], Asier Sáez-Cirión [4], Roger Le Grand [1] & Mariangela Cavarelli [1] ✉

Achieving durable viral remission without antiretroviral therapy (ART) remains a central challenge in HIV-1 cure research. Using a pathogenic SIV model in male cynomolgus macaques, we investigated mucosal and systemic immune features associated with post-treatment control (PTC). Chronic SIV infection disrupts intestinal macrophage homeostasis, skewing the compartment toward a CX3CR1low inflammatory phenotype marked by increased expression of costimulatory and homing markers. This polarization is associated with mucosal CD4+ T cell depletion, elevated neutrophil activation, and systemic cytokine induction. Non-controllers exhibit a similar inflammatory profile. In contrast, PTCs maintain CX3CR1high macrophages, preserve regulatory CD4 + T cells, and exhibit attenuated mucosal and systemic immune activation, resembling uninfected animals. CX3CR1high macrophage abundance inversely correlates with viral burden, T cell activation, and pro-inflammatory cytokines, suggesting a potential role in post-treatment control. These findings identify CX3CR1-expressing intestinal macrophages as a potential biomarker of mucosal immune regulation and highlight their relevance as targets in HIV cure strategies.

Despite the efficacy of antiretroviral therapy (ART) in suppressing HIV-1 replication, viral rebound typically occurs within weeks after treatment interruption due to the persistence of long-lived latent reservoirs and, potentially, residual replication in pharmacological sanctuaries[1–5]. A small subset of individuals, known as post-treatment controllers (PTCs), maintain durable viral replication control after ART cessation, offering critical insights into the mechanisms required for sustained remission[4–11]. However, the immunological determinants underlying

this phenotype remain incompletely understood, particularly at mucosal sites, where much of the immune damage occurs during HIV/SIV infection.

The gastrointestinal tract is a major site of early CD4 + T cell depletion, viral replication, and chronic inflammation during HIV-1/SIV infection[12–15]. Damage to the mucosal barrier contributes to microbial translocation and systemic immune activation, both of which are tightly linked to disease progression and are only partially resolved by

[1]Université Paris-Saclay, Inserm, CEA; Immunological diseases, microbiology and innovative therapies (IDMIT/UMRS1184), Fontenay-aux-Roses & Le Kremlin-Bicêtre, France. [2]Université Paris Cité; INSERM, U1016; CNRS, Paris, France. [3]Université d'Orléans, CHU d'Orléans, Orléans, France. [4]Institut Pasteur, Université Paris Cité, Viral Reservoirs and Immune Control Unit, Paris, France. [5]Present address: Commissariat à l'Energie Atomique et aux Energies Alternatives (CEA), Institut National de Recherche pour l'Agriculture, l'Alimentation et l'Environnement (INRAE), Département Médicaments et Technologies pour la Santé, Service d'Ingénierie Moléculaire pour la Santé (SIMoS), Université de Paris-Saclay, Gif-sur-Yvette, France. ✉e-mail: mariangela.cavarelli@cea.fr

ART[13,16–20]. Maintaining immune homeostasis in the gut requires finely tuned coordination between epithelial cells, lymphocytes, and mononuclear phagocytes (MNPs), particularly macrophages (Mφ) and dendritic cells, that orchestrate antigen sampling, barrier repair, and regulation of adaptive immunity[21–24].

Among MNPs, CX3CR1+ Mφ have emerged as key regulators of mucosal tolerance and tissue integrity[21,25–28]. In healthy tissues, CX3CR1high Mφ are associated with epithelial regeneration and anti-inflammatory functions, whereas CX3CR1low Mφ exhibit impaired maturation and inflammatory activity[29–33]. In pathogenic SIV infection, we have demonstrated that this balance is rapidly disrupted, with a decrease of CX3CR1high cells and the concomitant appearance of inflammatory CX3CR1low subsets as early as three days post-infection[34]. However, whether these perturbations persist during the chronic phase and ART interruption remains unknown. Furthermore, whether Mφ composition differs between PTCs and NCs, and whether it contributes to post-treatment immune equilibrium, has not been addressed.

Beyond the mucosa, chronic immune activation extends to lymphoid and systemic compartments and is a hallmark of HIV/SIV pathogenesis[35–37]. Elevated T cell activation, pro-inflammatory cytokines, and dysregulated myeloid populations contribute to ongoing tissue damage and poor clinical outcomes despite viral suppression[38,39]. Prior work has linked the activation of monocytes and neutrophils to systemic inflammation in HIV-1 infection[40–46], but the crosstalk between Mφ polarization and downstream immune consequences in the post-treatment setting remains poorly defined.

This work is part of the *pVISCONTI* study, which involves the nonhuman primate (NHP) model of SIVmac251 infection in male cynomolgus macaques undergoing structured ART and analytical treatment interruption (ATI). The original study showed that early ART facilitates post-treatment control by enhancing long-lived memory CD8+ T cells with potent recall and viral suppressive functions[47]. Here, we extend this analysis to investigate the cellular and molecular features of mucosal immune homeostasis in the post-treatment setting, aiming to define the contribution of innate immune polarization to durable post-ART viral control.

We hypothesized that the restoration of CX3CR1high intestinal Mφ promotes mucosal immune regulation, limits systemic inflammation, and contributes to a PTC status after ART cessation. To test this, we performed deep phenotyping of intestinal and lymphoid immune populations, along with plasma cytokine profiling. We identify the maintenance of CX3CR1high Mφ as a distinguishing feature of PTCs, tightly associated with reduced neutrophil activation, lower pro-inflammatory cytokine levels, and restoration of mucosal CD4+ T cells. These findings highlight intestinal Mφ polarization as a central determinant of post-treatment immune outcomes and a promising target for interventions aiming to achieve functional cure.

## Results

### Identification of post-treatment controllers among SIV-infected cynomolgus macaques

To investigate the immunological correlates of post-treatment control, 25 adult cynomolgus macaques (CM) were intravenously infected with SIVmac251 and subsequently divided into three groups (Fig. 1A): chronic animals that remained untreated until euthanasia (SIV +, $n = 12$, Gr.2), early-treated animals that received ART at 4 weeks post-infection ($n = 9$, Gr.3), and late-treated animals that began ART at 24 weeks post-infection ($n = 4$, Gr.4). ART was maintained for 2 years in both treated groups, followed by analytical treatment interruption (ATI). Animals were monitored for at least 6 months post-ATI. Additionally, 12 uninfected adult CM (SIV-, Gr.1) were included as negative controls.

All infected animals exhibited a peak in plasma viral load (pVL) by day 10 post-infection (Fig. 1B−D). Among the SIV+ group, three animals

(SIV + 4, 7, and 12) showed transient pVL declines to or below 400 copies/mL at least once during the follow-up and were classified as "natural controllers" (Fig. 1B); they remained untreated and, consistent with the predefined study design based on ART exposure and post-ATI outcome, were included in the SIV+ group for all analyses and are marked with star symbols in the figures. ART effectively suppressed pVL with occasional blips in both early- and late-treated groups (Fig. 1C, D). Following ATI, animals were classified as PTC if they achieved at least one pVL measurement below 400 RNA copies/ml after viral rebound, or as non-controllers (NC) if pVL remained persistently above this threshold (Fig. 1C, D). One animal (PTC9), which received late ART, exhibited spontaneous control of viremia prior to ART initiation and showed sustained low-level viremia after ATI; based on its virological profile, this animal was included in the PTC group. Early ART initiation resulted in a higher proportion of PTCs (8/9) than late ART (2/4) (Fig. 1C, D), consistent with findings from the larger pVISCONTI reference cohort[47].

At necropsy, pVL was significantly lower in PTCs than in SIV+ and NC animals (Fig. 1E), despite comparable cumulative viral exposure before ART initiation (Fig. 1F). Blood CD4 + T cell frequencies were restored to SIV− level in PTCs but depleted in SIV+ and NC animals (Fig. 1G), and inversely correlated with pVL (Fig. 1H). Cell-associated SIV DNA was significantly lower in PTCs relative to SIV+ animals in peripheral blood mononuclear cells (PBMCS; $q = 0.0050$), lymph node mononuclear cells (LNMC; $q = 0.0003$), and sigmoid tissues ($q = 0.0006$) (Fig. 1I–K). SIV-DNA levels in PBMCS were also significantly lower in PTCs than in NCs ($q = 0.0057$), suggesting a link between reservoir control and post-treatment viral suppression.

### Impact of SIV infection and ART interruption on intestinal myeloid cells

We next assessed whether post-treatment control was associated with preservation of gut myeloid populations. Flow cytometric analysis on *lamina propria* mononuclear cells (LPMCs) from sigmoid colon obtained at necropsy showed that all infected groups exhibited reduced CD45+ cell frequencies compared to SIV− controls ($q = 0.02$ for SIV+ and PTC; $q = 0.0056$ for NC) (Supplementary Fig. 1A). Plasmacytoid dendritic cells (pDCs) were significantly reduced in both PTC and NC animals relative to SIV+ and SIV− groups ($q = 0.0364$ and $q = 0.0231$ for PTC vs SIV− and SIV+; $q = 0.0231$ and $q = 0.231$ for NC vs. SIV− and SIV +, respectively; Supplementary Fig. 1B). While total myeloid dendritic cell (DCs) frequencies were comparable (Supplementary Fig. 1C), tolerogenic CD103+ DCs were decreased in SIV+ animals ($q = 0.0327$ vs SIV-; Supplementary Fig. 1D), and CD103− DCs were reduced in PTCs ($q = 0.0105$ vs. SIV−; $q = 0.0026$ vs. SIV+; Supplementary Fig. 1E). Overall, Mφ abundance among CD45+ cells remained comparable between groups, except for one outlier in the SIV+ group (Supplementary Fig. 1F).

### SIV infection disrupts intestinal macrophage homeostasis and skews CX3CR1 expression

Although total Mφ frequencies remained stable, we next investigated whether the distribution of phenotypically distinct macrophage subsets was altered during chronic SIV infection. To establish how SIV infection alone perturbs intestinal macrophage homeostasis, this initial analysis was restricted to SIV− and SIV+ animals. This infection-focused comparison defines the baseline alterations in Mφ phenotype before introducing the effects of ART and post-treatment outcome examined in subsequent sections. CX3CR1+ Mφ, known to regulate mucosal immune surveillance, microbial containment, and epithelial repair, are critical to intestinal immune homeostasis. We therefore characterized *lamina propria* Mφ in SIV− and SIV+ animals, identifying them as Lineage− HLA-DR+ CD64+ cells (Supplementary Fig. 2A) and further stratifying them into phenotypic subsets based on expression of CD14 (monocyte derivation), CD11c (maturation state), and CX3CR1 expression.

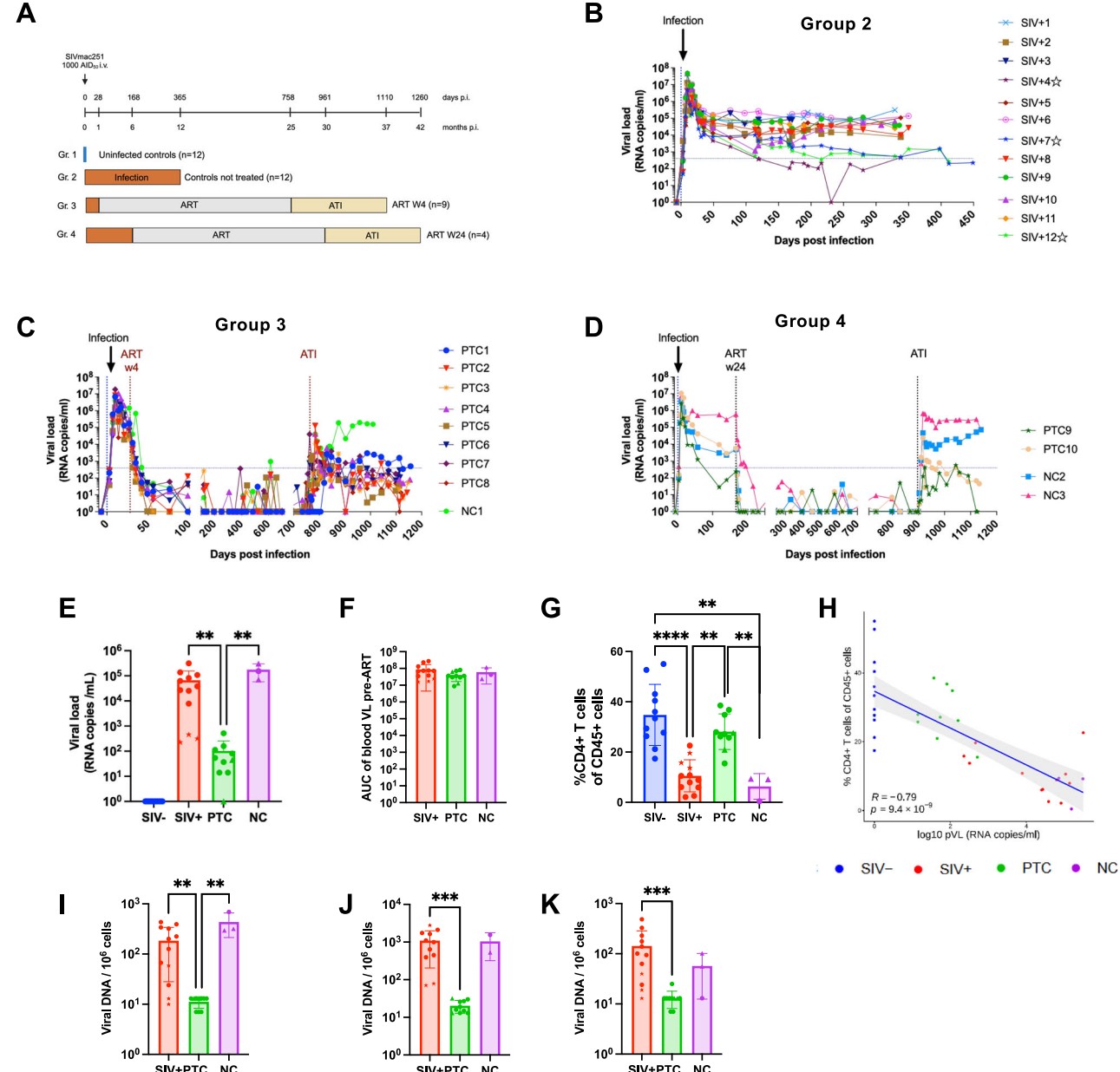

**Fig. 1 | Identification of post-treatment controllers following early or late ART initiation among SIV-infected cynomolgus macaques. A** Experimental design showing the study groups: SIV⁻ (Group 1; uninfected controls), SIV⁺ (Group 2; untreated chronic), early-treated animals receiving ART from week 4 post-infection (Group 3), and late-treated animals receiving ART from week 24 post-infection (Group 4). Both treated groups underwent analytical treatment interruption (ATI) after two years of ART and were classified as post-treatment controllers (PTC) or non-controllers (NC) based on virological outcomes. **B–D** Plasma viral load (pVL) kinetics in Group 2 (**A**), Group 3 (**B**), and Group 4 (**D**) animals following SIVmac251 infection and structured ART. Individual trajectories are shown; animals marked with a star in (B) are natural controllers. **E** pVL at necropsy in SIV⁻ ($n = 12$), SIV⁺ ($n = 12$), PTC ($n = 10$), and NC ($n = 3$) animals. **F** Cumulative pVL before ART initiation (area under the curve, AUC) in SIV⁺ ($n = 12$), PTC ($n = 10$), and NC ($n = 3$) animals. **G** CD4 + T cell frequencies at necropsy in blood in SIV⁻ ($n = 11$), SIV⁺ ($n = 12$), PTC ($n = 10$), and NC ($n = 3$) animals. **H** Spearman correlation between pVL and peripheral blood CD4⁺ T cell counts. **I–K** Quantification of SIV DNA in PBMCS (**I**), lymph node mononuclear cells (LMNCs, J), and the sigmoid colon (**K**) at necropsy in SIV⁺ ($n = 12$ for PBMCs and sigmoid colon and $n = 11$ for LMNCs), ($n = 12$), PTC ($n = 10$), and NC ($n = 3$ for PBMCs and sigmoid colon and $n = 2$ for LMNCs) animals. *Spearman correlation was two-sided. Kruskal–Wallis with Benjamini–Krieger–Yekutieli (FDR) correction; $q < 0.05$ (\*), $q < 0.01$ (\*\*), $q < 0.001$ (\*\*\*).* In the histograms, symbol shapes indicate the study group: circles represent SIV⁻ and SIV⁺ samples as well as PTCs and NCs from Group 3; stars represent natural controllers; triangles represent PTCs and NCs from Group 4. Symbol shapes and colour coding are consistent across all figures. Colours denote experimental groups: blue, SIV⁻; red, SIV⁺; green, PTC; magenta, NC. The results are presented as individual data points together with the mean ± standard deviation. Source data are provided as a Source Data file.

In SIV⁻ animals, Mφ were predominantly CD14⁻ CD11c⁻ CX3CR1^high (Fig. 2A–C, **left**) whereas in SIV⁺ animals, there was a marked shift toward CD14⁺ CD11c⁺ CX3CR1^low phenotypes ($p = 0.0005$, 0.0017, and <0.0001, respectively; Fig. 2A–C, **right**). This was further reflected by a significant decrease in the CX3CR1^high/CX3CR1^low Mφ ratio ($p < 0.001$; Fig. 2D). CX3CR1 positive and negative gating was set using the fluorescence minus one (FMO) method (Fig. 2E).

To further dissect this phenotype, we examined CX3CR1 expression by mature (CD11c⁻) and immature (CD11c⁺) cells within CD14⁻ and CD14⁺ subsets. In SIV⁻ animals, mature CD11c⁻ Mφ maintained high

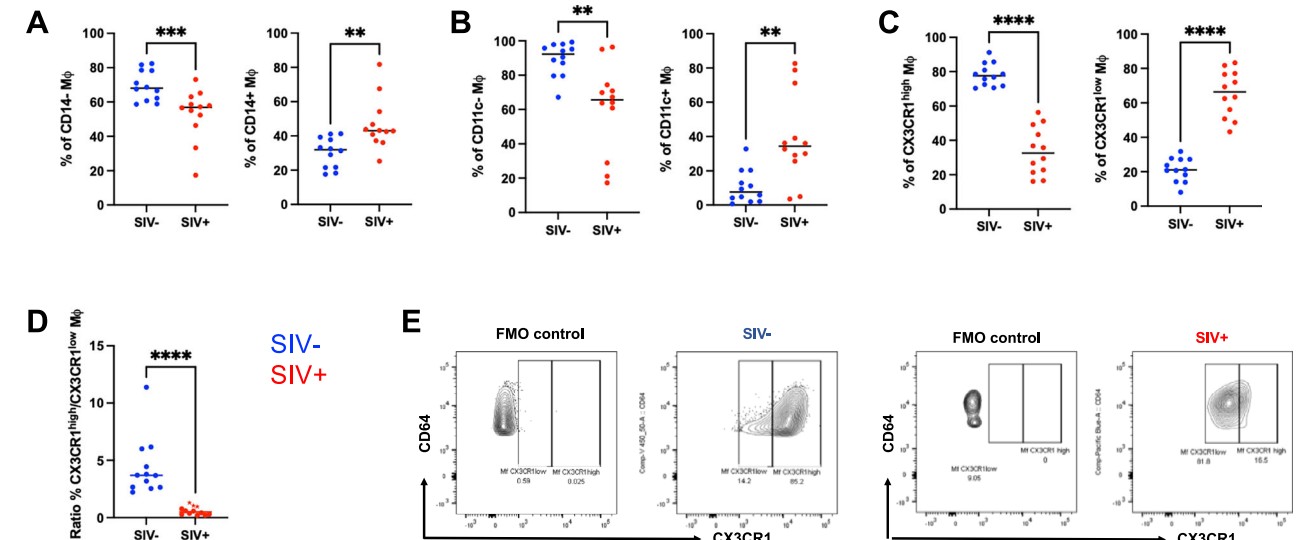

**Fig. 2 | Phenotypic characterization of intestinal macrophages in SIV− and SIV+ animals. A−C** Frequencies of CD14⁻ and CD14⁺ (**A**), CD11c⁻ and CD11c⁺ (**B**), and CX3CR1$^{high}$ and CX3CR1$^{low}$ (**C**) subsets among total macrophages in the sigmoid colon in SIV⁻ ($n = 12$), SIV⁺ ($n = 12$) animals. **D** Ratio of CX3CR1$^{high}$ to CX3CR1$^{low}$ macrophages SIV⁻ ($n = 12$), SIV⁺ ($n = 12$) animals. The results are presented as individual data points together with the mean ± standard deviation. *Mann−Whitney two-sided U test; p < 0.05 (\*), p < 0.01 (\*\*), p < 0.001 (\*\*\*).* **E** Representative flow cytometry plots of CX3CR1 expression and CD64 levels in SIV⁻ and SIV⁺ animals compared to fluorescence-minus-one (FMO) controls. Source data are provided as a Source Data file.

CX3CR1 levels, while immature CD11c⁺ Mφ also retained a partial CX3CR1$^{high}$ phenotype (CD11c⁺ CX3CR1$^{high}$). This pattern was consistent across CD14⁻ and CD14⁺ populations (Supplementary Fig. 3A, B, **left**). In contrast, SIV⁺ animals exhibited a global skewing toward CX3CR1$^{low}$ phenotypes, irrespective of maturation and CD14 status (Supplementary Fig. 3A, B, **right**).

Together, these results indicate that SIV infection disrupts intestinal macrophage homeostasis, characterized by a loss of mature CX3CR1$^{high}$ subsets and the expansion of CX3CR1$^{low}$ cells.

## Post-treatment control is associated with the restoration of CX3CR1⁺ macrophage balance

To evaluate whether the perturbations in Mφ phenotype observed in chronic infection are reversed in PTCs, we compared CX3CR1 expression in total, CD14⁺/⁻, and CD11c⁺/⁻ Mφ across SIV⁻, SIV⁺, PTC, and NC groups.

PTC animals displayed significantly higher frequencies of CX3CR1$^{high}$ Mφ and reduced CX3CR1$^{low}$ subsets compared to SIV⁺ and NC animals, closely resembling the profile of SIV- animals (Fig. 3A, B). Conversely, NC animals maintained a CX3CR1 distribution similar to chronically infected animals. Frequencies of CX3CR1$^{high}$ Mφ inversely correlated with pVL and PBMCS SIV DNA ($r = -0.67$ and 0.73; $p < 0.0001$), while CX3CR1$^{low}$ Mφ correlated positively ($r = 0.66$ and 0.73; $p < 0.0001$) (Fig. 3C, D; and Supplementary Fig. 4A, B).

This polarization profile was consistent across CD14⁻ and CD14⁺ (Fig. 3E–J), as well as mature (CD11c⁻) and immature (CD11c⁺) subsets (Supplementary Fig. 4C, D). Notably, both CD14⁺ and CD14⁻ CX3CR1$^{high}$ subsets negatively correlated with pVL ($r = -0.76$ and −0.8; $p < 0.0001$), while CX3CR1$^{low}$ counterparts showed strong positive associations ($r = 0.76$ and 0.81; $p < 0.0001$; Fig. 3G, H, K, L), highlighting the robustness of this polarization as a correlate of viral control. In line with this, spontaneous controllers exhibited the highest frequencies of CX3CR1$^{high}$ Mφ and the lowest frequencies of CX3CR1$^{low}$ Mφ among SIV⁺ animals, displaying a profile closer to that observed in PTCs and SIV-animals across both total and CD14 Mφ subsets (Fig. 3A, B, E, F, I, J).

## CX3CR1$^{high}$ and CX3CR1$^{low}$ macrophages exhibit distinct inflammatory phenotypes

To assess the functional polarization of CX3CR1-defined Mφ subsets, we analyzed the expression of costimulatory (CD40, CD80, CD86), maturation (CD83), and homing (α4β7, CCR5, CCR9, CD62L) markers within CX3CR1$^{high}$ and CX3CR1$^{low}$ subsets (gating in Supplementary Fig. 2C).

CX3CR1$^{low}$ Mφ from SIV⁺ animals exhibited increased expression of CD40 ($q = 0.0102$), CD83 ($q = 0.0119$), CD86 ($q = 0.0002$), and α4β7 ($q = 0.0397$) compared to PTC or SIV⁻ animals, consistent with an activated, pro-inflammatory, and gut-homing phenotype (Fig. 4A–D). In contrast, CD69 expression by CX3CR1$^{high}$ Mφ was reduced in infected animals, with a significant decrease in the SIV⁺ group compared to SIV⁻ controls ($p = 0.0071$) (Fig. 4E), while CCR5 expression was elevated in PTC animals relative to SIV+ animals ($q = 0.0023$) (Fig. 4F), consistent with a tissue-resident, homeostatic profile. No significant differences were observed for CCR9, CD80, or CD62L across groups (Supplementary Fig. 5).

## Persistent intestinal immune activation and CD4 + T cell dysfunction associated with CX3CR1$^{low}$ macrophages in SIV-infected animals

In line with the gut damage seen in HIV/SIV infection, both SIV⁺ and NC animals displayed profound CD4⁺ T cell depletion within the sigmoid colon (Fig. 5A; gating in Supplementary Fig. 6), with strong inverse correlation to pVL ($r = -0.83$; $p < 0.0001$) (Supplementary Fig. 7A). Despite viral control, PTCs still had lower CD4⁺ T cell frequencies than SIV- animals, indicating incomplete immune reconstitution (Fig. 5A).

NC animals showed reduced central memory (CM) CD4⁺ T cells compared to both SIV- and PTC animals (Fig. 5B), while both SIV⁺ and NC groups exhibited an increased frequency of stem cell-like memory (Tscm) relative to PTCs (Fig. 5C). This shift in memory subset distribution was accompanied by elevated PD-1 expression in SIV⁺ and NC animals (Fig. 5D). In line with these findings, pVL negatively correlated with CM CD4⁺ T cells ($r = -0.48$; $p = 0.0037$), and positively with Tscm CD4⁺ T cells ($r = 0.36$; $p = 0.035$) and PD-1 expression ($r = 0.47$; $p = 0.004$) (Supplementary Fig. 7A–D).

Evaluation of additional activation markers (Supplementary Fig. 6A) revealed that CD4⁺ T cells from PTC animals expressed lower HLA-DR, Ki-67, and α4β7 (Fig. 5E–G), and higher CD69 (Fig. 5H), relative to SIV⁺ or NC animals. Ki-67 expression positively correlated with pVL ($r = 0.36$; $p = 0.035$) (Supplementary Fig. 7E), supporting the association between viral burden and T cell proliferation. Spontaneous

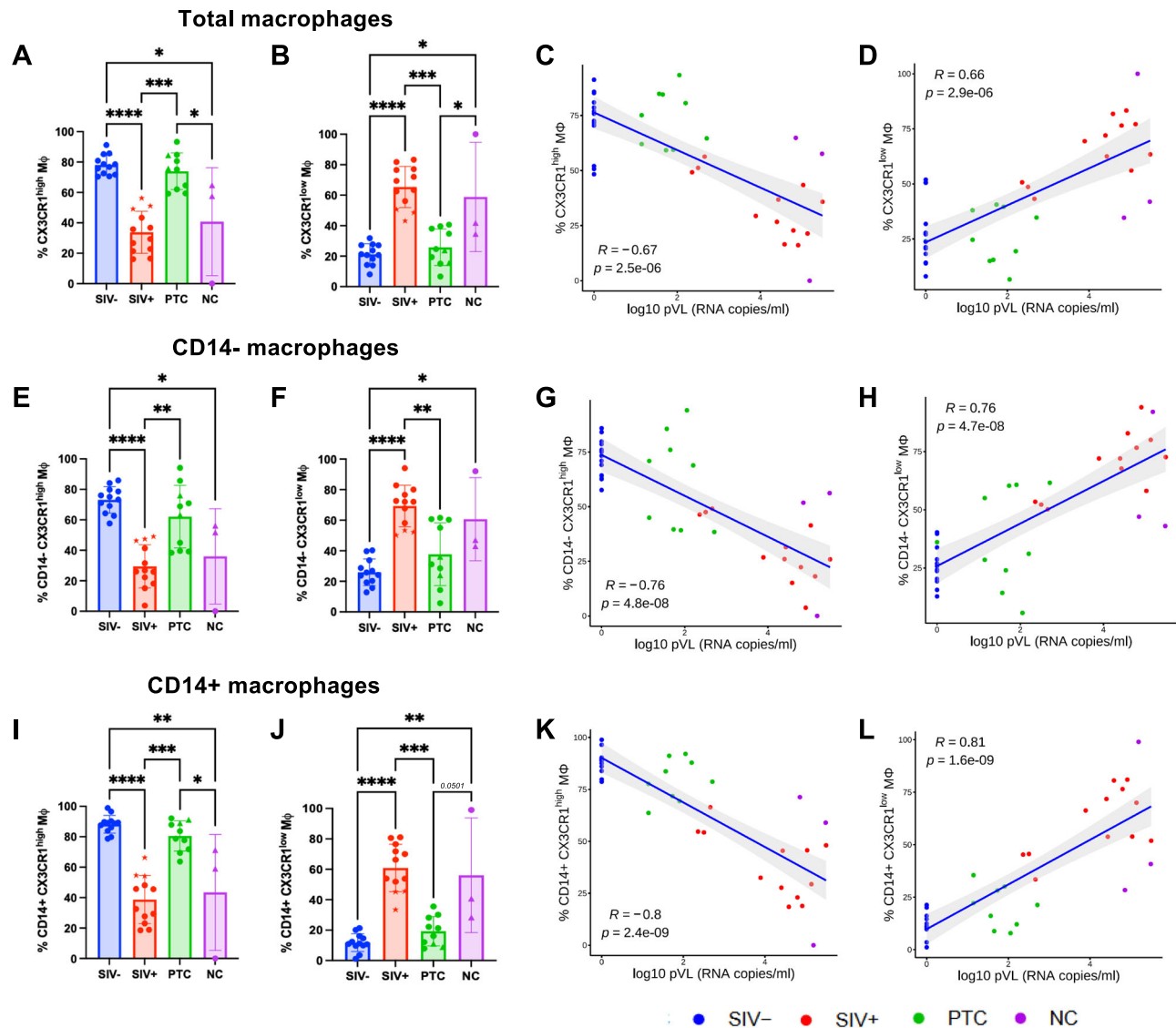

**Fig. 3 | CX3CR1^high macrophages are preserved in post-treatment controllers and inversely correlate with plasma viral load. A, B** Frequency of CX3CR1^high and CX3CR1^low macrophages among total sigmoid macrophages (M_Φ) in SIV⁻ (*n* = 12), SIV⁺ (*n* = 12), PTC (*n* = 10), and NC (*n* = 3) animals. **C, D** Spearman correlation between plasma viral load (pVL) and the frequency of CX3CR1^high (**C**) or CX3CR1^low (**D**) macrophages among total macrophages. **E, F** Frequencies of CD14⁻CX3CR1^high and CD14⁻CX3CR1^low macrophages among total macrophages in SIV⁻ (*n* = 12), SIV⁺ (*n* = 12), PTC (*n* = 10), and NC (*n* = 3) animals. **G, H** Spearman correlation between pVL and CD14⁻CX3CR1^high (**G**) or CD14⁻CX3CR1^low (**H**) macrophages. **I, J** Frequencies

of CD14⁺CX3CR1^high and CD14⁺CX3CR1^low macrophages among total macrophages in SIV⁻ (*n* = 12), SIV⁺ (*n* = 12), PTC (*n* = 10), and NC (*n* = 3) animals. **K, L** Spearman correlation between pVL and CD14⁺CX3CR1^high (**M**) or CD14⁺CX3CR1^low (**N**) macrophages. All spearman correlation were two-sided. *Kruskal–Wallis with Benjamini–Krieger–Yekutieli (FDR) correction; q < 0.05 (\*), q < 0.01 (\*\*), q < 0.001 (\*\*\*).* Spearman's rank correlation is used for correlation analyses. The results are presented as individual data points together with the mean ± standard deviation. Source data are provided as a Source Data file.

controllers displayed CD4⁺ T-cell frequencies within the upper range of SIV⁺ animals (Fig. 5A), although for other subsets their values overlapped with the broader SIV⁺ distribution (Fig. 5B–K).

Functional analysis of CD4⁺ T cell subsets by intracellular cytokine staining (gating in Supplementary Fig. 6B, C) revealed increased Th1 cells (IFN-γ⁺ IL-17⁻) in SIV⁺ animals compared to SIV⁻ controls (*p* = 0.0259), and increased Th17 cells (IFN-γ⁻ IL-17⁺) in NC animals relative to SIV⁺ (*p* = 0.0111) (Fig. 5I, J). Regulatory T cells (Treg) were significantly reduced in both SIV⁺ and NC animals compared to SIV⁻ and PTC groups (Fig. 5K). Th1 cell frequency positively correlated with pVL (*r* = 0.37; *p* = 0.031) while Treg frequency showed an inverse correlation (r = −0.55; *p* < 0.0001) (Supplementary Fig. 7F, G), supporting their opposing roles in mucosal immune balance.

Given the central role of macrophage-T cell crosstalk in maintaining gut immune homeostasis, we examined correlations between M_Φ subsets and CD4⁺ T cell phenotypes. CX3CR1^high M_Φ were positively associated with CD4⁺ Treg (r = 0.54; *p* = 0.0012) and Th17 cells (*r* = 0.47; *p* = 0.0068) (Fig. 6A, C), whereas CX3CR1^low M_Φ showed negative correlations with these subsets (*r* = −0.54; *p* = 0.0013 and r = −0.46; *p* = 0.0073, respectively) (Fig. 6B, D). Conversely, Th1 cells displayed a trend toward a negative association with CX3CR1^high M_Φ (*r* = −0.34; *p* = 0.052) (Fig. 6E) and toward a positive association with CX3CR1^low M_Φ (*r* = 0.32; *p* = 0.066) (Fig. 6F). These results support a model in which CX3CR1^high M_Φ promotes an anti-inflammatory environment by promoting CD4+ Treg induction and sustaining mucosal barrier integrity through Th17 cell induction, whereas CX3CR1^low M_Φ

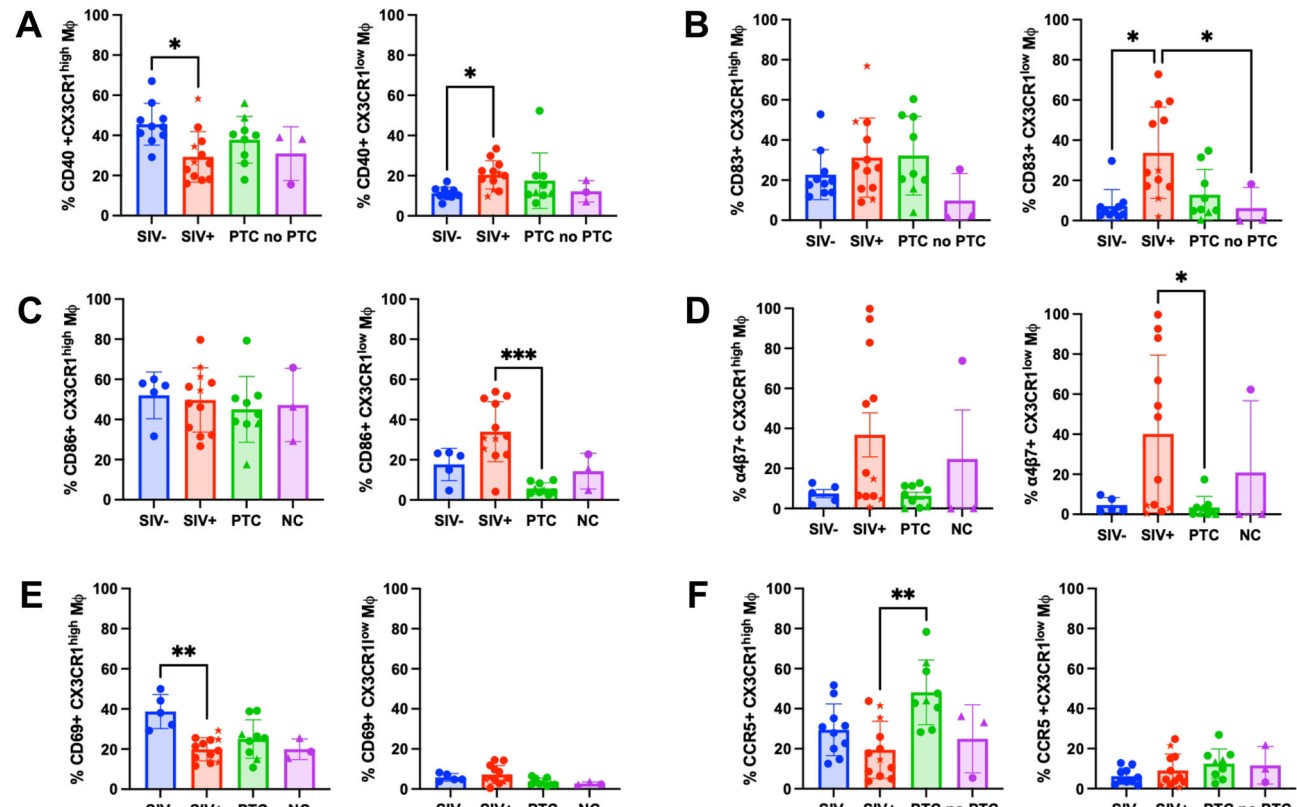

**Fig. 4 | CX3CR1^high and CX3CR1^low macrophages exhibit distinct activation and homing phenotypes in the intestinal mucosa.** Frequencies of CX3CR1^high and CX3CR1^low macrophages expressing CD40 (**A**), CD83 (**B**), CD86 (**C**), α4β7 (**D**), CD69 (**E**), and CCR5 (**F**) were measured in the sigmoid colon of SIV⁻ (*n* = 10 for CD40, CD83 and CCR5, *n* = 5 for CD86, α4β7 and CD69), SIV⁺ (*n* = 12), PTC (*n* = 9), and NC (*n* = 3) animals. *Kruskal–Wallis with Benjamini–Krieger–Yekutieli (FDR) correction; q < 0.05 (\*), q < 0.01 (\*\*), q < 0.001 (\*\*\*).* The results are presented as individual data points together with the mean ± standard deviation. Source data are provided as a Source Data file.

tend to drive Th1-skewed inflammation and mucosal dysfunction. We next examined whether CX3CR1-defined macrophage subsets were associated with additional CD4⁺ memory T-cell compartments. No significant correlation was observed between CX3CR1^high Mφ frequencies and CM CD4⁺ T cells. A statistically significant inverse correlation was detected between CX3CR1^high Mφ and frequency of Tscm CD4⁺ T cells (R = −0.59, *p* = 0.00029), and a direct correlation with CX3CR1^low Mφ (r = 0.59, *p* = 0.00032; Fig. 6G, H). Total CD4⁺ T cell frequency also positively correlated with CX3CR1^high Mφ (r = 0.54; *p* = 0.0014), and negatively with CX3CR1^low Mφ (r = −0.54; *p* = 0.0016) (Fig. 6I), suggesting that CX3CR1^high Mφ contributes to the preservation of mucosal CD4⁺ T cell pools.

Finally, we assessed whether macrophage polarization states were associated with the activation profile of intestinal CD4⁺ T cells. CX3CR1^low Mφ were positively correlated with activation and homing markers on CD4⁺ T cells, including HLA-DR (r = 0.47; *p* = 0.006), Ki-67 (r = 0.54; *p* = 0.0012), PD-1 (r = 0.62; *p* = 0.0001), and α4β7 (r = 0.45; *p* = 0.01) (Fig. 6I). Conversely, these markers were inversely associated with CX3CR1^high Mφ (Fig. 6I), consistent with their homeostatic role. Similar patterns were observed in the CD14⁺ subsets of CX3CR1^low and CX3CR1^high Mφ macrophage compartments (Supplementary Fig. 8), reinforcing the role of macrophage polarization in shaping mucosal T cell responses.

**Enhanced neutrophil activation is associated with inflammatory CX3CR1^low macrophages**

Neutrophils are increasingly recognized as key drivers of mucosal inflammation in HIV/SIV infection, yet their interplay with macrophage populations remains poorly defined. To address this, we profiled colonic neutrophil subsets in available animals (gating in Supplementary Fig. 9A).

Total, mature, and immature neutrophil frequencies did not differ significantly among groups (Fig. 7A–C); however, neutrophil activation markers varied according to infection status. In viremic animals (SIV⁺ and NCs), the proportion of CD11b^high neutrophils was significantly elevated, with a corresponding decrease in CD11b^low cells across total, immature, and mature subsets (CD11b^high: *q* = 0.0226–0.0434 SIV⁻ vs SIV + ; *q* = 0.0030–0.0041 SIV⁻ vs NC; CD11b^low: *q* = 0.0215–0.0402 SIV⁻ vs SIV⁺; *q* = 0.0027–0.0045 SIV⁻ vs NC) (Fig. 7D–I), consistent with a more activated neutrophil phenotype. CD32a expression was also markedly increased in total and immature neutrophils from SIV+ animals (total: *q* = 0.0011–0.0375; immature: *q* = 0.0010–0.0353 SIV⁺ vs SIV⁻, PTCs, and NCs) (Fig. 7J, K), but not in the mature subset (Fig. 7L). By contrast, expression of CD64, HLA-DR, or PD-L1 remained unchanged across groups (Supplementary Fig. 9B–D).

The pVL positively correlated with both total and immature neutrophil frequencies (Supplementary Fig. 10A, B) as well as with CD11b^high (Supplementary Fig. 10C, D) and CD32a⁺ primed and activated neutrophil subsets (Supplementary Fig. 10E, F), further supporting an association between viral burden and neutrophil activation. Notably, CX3CR1^low Mφ frequencies strongly correlated with CD66⁺CD32a⁺ neutrophils across all compartments (total, immature, and mature; Fig. 7M and Supplementary Fig. 10I), suggesting coordinated activation of gut myeloid populations during persistent SIV infection. These findings implicate neutrophil–macrophage crosstalk as a potential amplifier of gut inflammation and viral persistence.

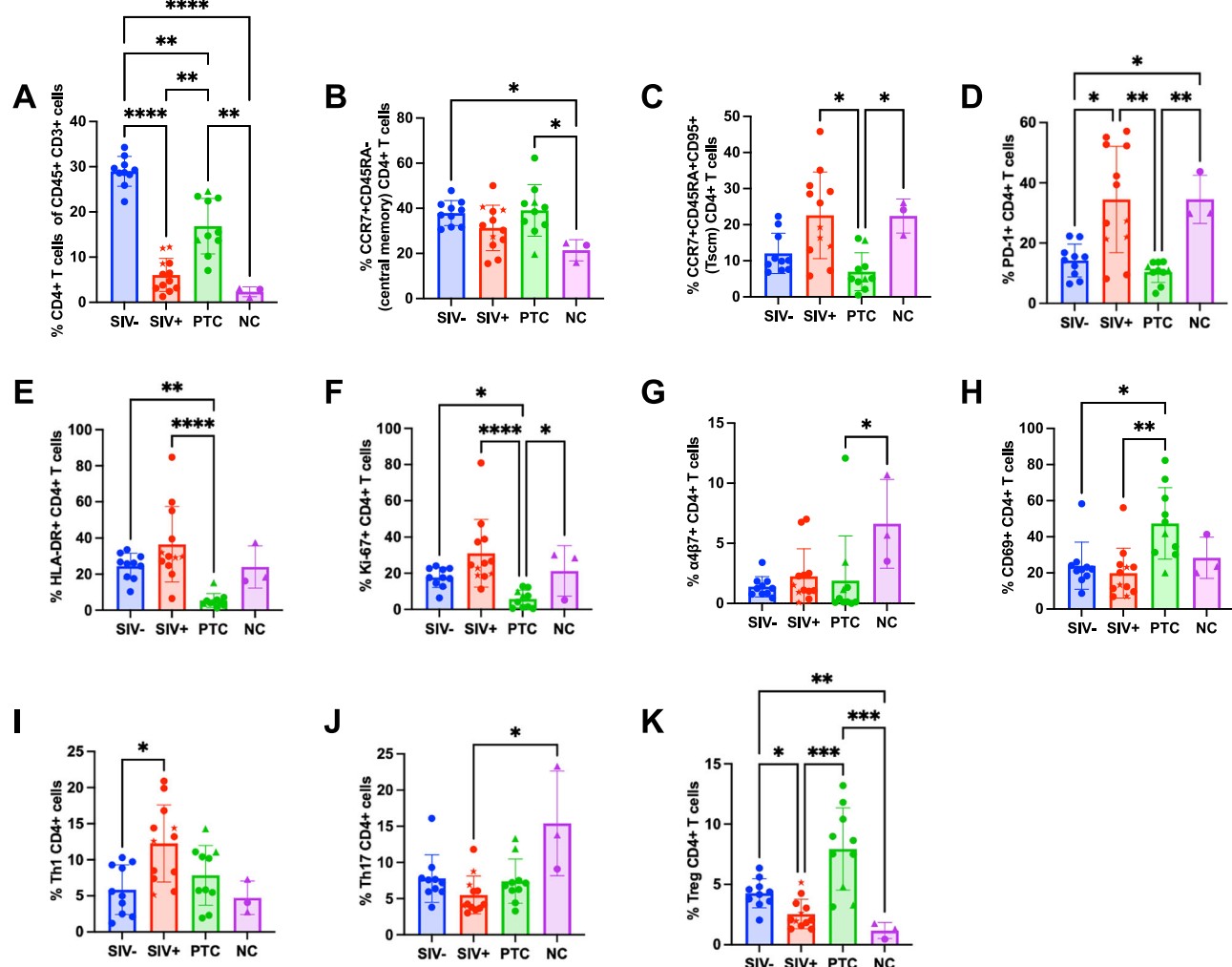

**Fig. 5 | Increased activation and Th1 polarization of intestinal CD4⁺ T cells in viremic animals. A** Frequencies of total CD4⁺ T cells among total CD45⁺ CD3+ cells in the sigmoid colon. **B**, **C** Frequencies of CD4⁺ T cell memory subsets: central memory (CCR7⁺CD45RA⁻; B) and Tscm (CCR7⁺CD45RA⁺CD95⁺; C). **D−H** Frequency of PD-1⁺ (**D**), HLA-DR⁺ (**E**), Ki-67⁺ (**F**), α4β7⁺ (**G**), and CD69⁺ (**H**) CD4⁺ T cells. **I−K** Proportions of Th1 (IFN-γ⁺ IL-17⁻; **I**), Th17 (IFN-γ⁻ IL-17⁺; **J**), and regulatory T cells (CD25⁺FOXP3⁺CD127⁻; **K**) among colonic CD4⁺ T cells. All these data were measured in sigmoid colon of SIV⁻ ($n = 10$), SIV⁺ ($n = 12$), PTC ($n = 10$), and NC ($n = 3$) animals. *Kruskal–Wallis with Benjamini–Krieger–Yekutieli (FDR) correction; q < 0.05 (\*), q < 0.01 (\*\*), q < 0.001 (\*\*\*).* The results are presented as individual data points together with the mean ± standard deviation. Source data are provided as a Source Data file.

## PCA-based immune profiling distinguishes post-treatment controllers from non-controllers at the mucosal level

To integrate the multiparametric immune data obtained from the sigmoid colon, we performed principal component analyses (PCA) on innate immune cells (including monocytes, mDC, pDC, macrophages, and neutrophils), and CD4⁺ T cells (Supplementary Fig. 11). The PCA on innate immune cells (Supplementary Fig. 11A) revealed clear stratification of animals into three main clusters: SIV⁻, SIV⁺, and PTC. SIV⁻ animals were well separated from SIV⁺ animals across both dimensions, reflecting the extensive innate immune alterations associated with chronic infection. In contrast, PTC animals diverged from SIV⁻ primarily along the second principal component (dim 2), which accounted for only 9.2% of the total variance, suggesting a partial preservation or restoration of mucosal immune homeostasis. NC animals were distributed between PTC and SIV⁺ clusters, indicating that they share some innate immune features with viremic animals and others with PTCs. Variable contribution analysis (Supplementary Fig. 11B) showed that macrophage parameters, particularly CX3CR1 expression, were the major contributors to the variance along the first two principal components and were key drivers of group separation. The PCA focused on CD4⁺ T cells (Supplementary Fig. 11C) recapitulated these

patterns. SIV⁺ animals formed a distinct cluster, separated from both SIV- and PTC animals, and partially overlapping with NC, consistent with shared features of T cell activation and dysregulation. In contrast, PTC animals displayed variable positioning, with some clustering near SIV⁻ animals and others closer to NC/SIV⁺ groups, suggesting a partial normalization of the mucosal CD4⁺ T cell parameters. Correlation circle analysis (Supplementary Fig. 11D) revealed that activation markers on CD4⁺ T cells primarily contributed to the clustering of SIV⁺ and NC animals, while total CD4⁺ T cell frequencies and Treg abundance drove the clustering of SIV⁻ and PTC animals.

Together, these analyses indicate that, despite 2 years of ART initiated relatively early, NC animals retain a mucosal immune profile largely overlapping with that of viremic SIV⁺ animals, whereas PTC animals exhibit a distinct immune landscape associated with reduced inflammation and preserved homeostasis.

## Immune activation extends beyond the gut and is mitigated in PTCs

To determine whether mucosal immune activation reflects systemic changes, we analyzed the cellular phenotypes in colon-draining lymph nodes.

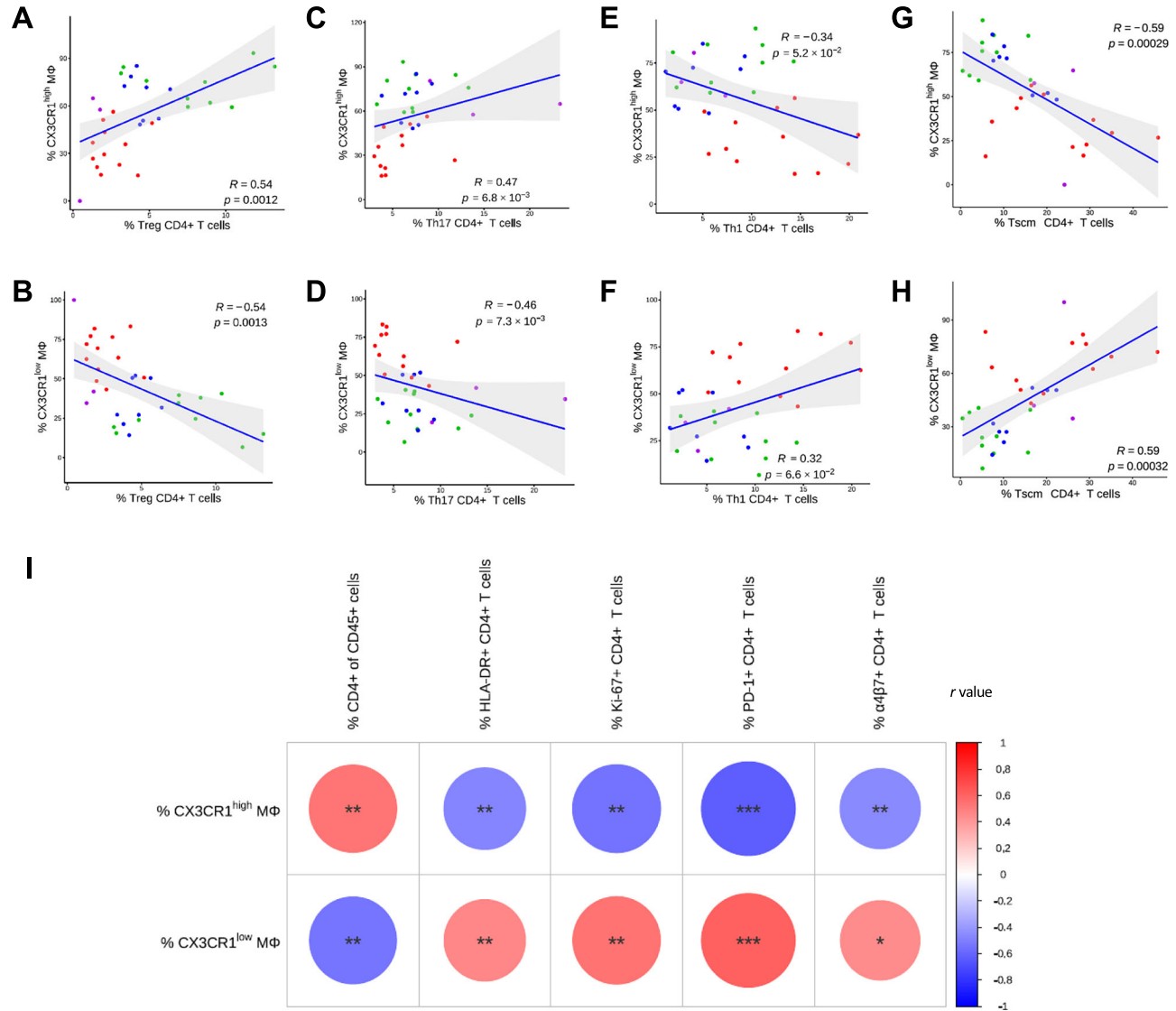

**Fig. 6 | CX3CR1$^{high}$ and CX3CR1$^{low}$ macrophages exhibit distinct correlations with intestinal CD4$^+$ T cell polarization and activation. A, B** Spearman correlations between total CX3CR1$^{high}$ (**A**) or CX3CR1$^{low}$ (**B**) macrophages and Treg CD4$^+$ T cell frequencies in the sigmoid colon. **C, D** Spearman correlations between CX3CR1$^{high}$ (**C**) or CX3CR1$^{low}$ (**D**) macrophages and Th17 CD4$^+$ T cells. **E, F** Spearman correlations between CX3CR1$^{high}$ (**E**) or CX3CR1$^{low}$ (**F**) macrophages and Th1 CD4$^+$ T cells. **G, H** Spearman correlations between CX3CR1$^{high}$ (**G**) or CX3CR1$^{low}$ (**G**) macrophages and Tscm CD4$^+$ T cells. Blue lines depict linear regression fits with 95% confidence intervals (gray shading). Confidence intervals are centered around the mean of the x value. **I** Heatmap summarizing Spearman correlation coefficients between CX3CR1$^{high}$ and CX3CR1$^{low}$ macrophages and multiple CD4$^+$ T cell parameters: total CD4$^+$, HLA-DR$^+$, Ki-67$^+$, PD-1$^+$, and α4β7$^+$ CD4$^+$ T cells. Spearman correlation coefficients (**R**) are represented by the color scale (red = positive correlations; blue = negative correlations) and by the size of the circles, which is proportional to the absolute value of R. Statistical significance is indicated by asterisks placed inside each circle (*$p$ < 0.05; **$p$ < 0.01; *$p$ < 0.001). All Spearman correlations were two-sided. No multiple comparison adjustment of the $p$ value of the Spearman correlations was made. Source data are provided as a Source Data file.

We found a significant accumulation of Mφ among CD45$^+$ cells in SIV$^+$ animals (q = 0.0007), with a similar trend in NCs (q = 0.0247) (Fig. 8A), indicating an inflammatory response in lymphoid tissues similar to that observed in the colon. This expansion was characterized by a skewing toward CX3CR1$^{low}$ Mφ and a reduction in CX3CR1$^{high}$ subsets in SIV$^+$ animals (q = 0.0154 and q = 0.0189, respectively, SIV$^+$ vs. PTC and SIV$^-$) (Fig. 8B). CX3CR1$^{low}$ Mφ in SIV$^+$ animals also expressed higher levels of CD86 and CD80, particularly among both CD11c$^+$ (immature) and CD11c$^-$ (mature) populations (Fig. 8C, D), consistent with an activated antigen-presenting cell phenotype.

This myeloid activation was accompanied by profound CD4$^+$ T cell alterations. Indeed, CD4$^+$ T cell frequencies were significantly reduced in both SIV$^+$ and NC animals compared to SIV- and PTCs (q = 0.0019 for all comparisons) (Fig. 8E). Remaining CD4$^+$ T cells in these groups

expressed elevated levels of PD-1 (q = 0.0153 in SIV$^+$ vs PTC; q = 0.0471 in NC vs PTC), HLA-DR (q = 0.0382 for all comparisons), and Ki-67 (q = 0.0224 in SIV$^+$ vs PTC) (Fig. 8F–H), indicating persistent activation and exhaustion. In contrast, PTC animals showed a lymphoid immune profile similar to uninfected animals, with reduced myeloid activation and preservation of quiescent CD4$^+$ T cell phenotypes.

Paired intestine–lymph node analyses revealed no significant correlation for the frequencies of total and CX3CR1$^{high}$ or CX3CR1$^{low}$ Mφ (Supplementary Fig. 12), indicating that the extent of CX3CR1 polarization is regulated locally within tissue. In contrast, CX3CR1 subsets expressing activation markers (CD80+, CD83+, CD86$^+$+), and/or CCR5 and α4β7, on either total Mφ (Supplementary Fig. 12) or CD11c+/CD11c- cells (Fig. 8I–L) displayed significant positive correlations between the two sites, suggesting that inflammatory

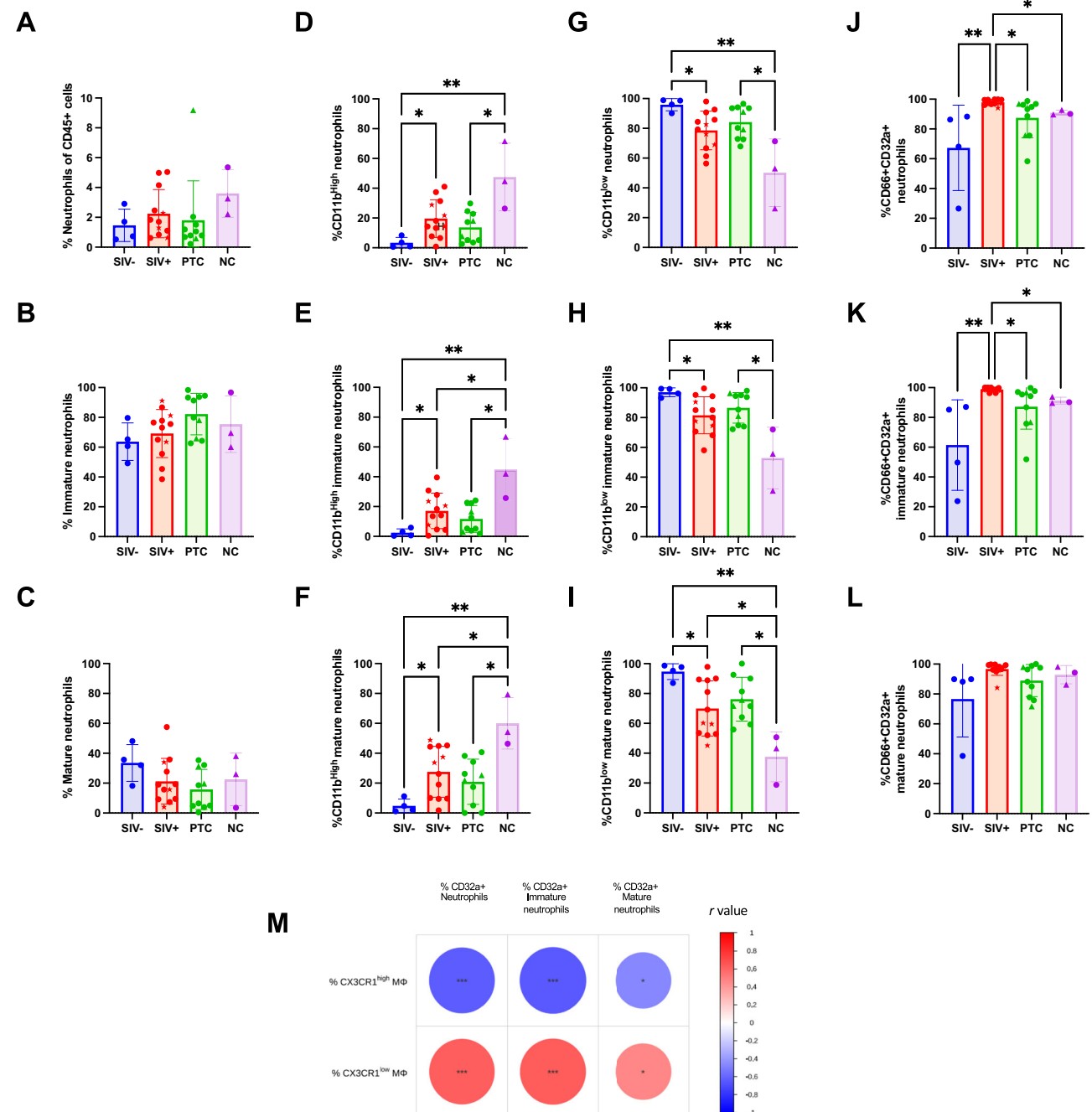

**Fig. 7 | Enhanced neutrophil activation is associated with pro-inflammatory CX3CR1^low macrophages in SIV-infected animals. A–C** Frequencies of total, immature (CD10⁻CD101⁺), and mature (CD10⁺CD101⁺) neutrophils among CD45⁺ cells in the sigmoid colon. **D–F** Proportions of CD11b^high neutrophils within total (D), immature (E), and mature (F) neutrophil subsets. **G–I** Proportions of CD11b^low neutrophils within total (**G**), immature (**H**), and mature (**I**) subsets. **J–L** Expression of CD32a in total (**J**), immature (**K**), and mature (**L**) neutrophils. All these data were measured in the sigmoid colon of SIV⁻ (*n* = 4), SIV⁺ (*n* = 12), PTC (*n* = 10), and NC (*n* = 3) animals *Kruskal–Wallis with Benjamini–Krieger–Yekutieli (FDR) correction; q < 0.05 (\*), q < 0.01 (\*\*), q < 0.001 (\*\*\*).* Spearman's rank correlation was used for

correlation analyses. The results are presented as individual data points together with the mean ± standard deviation. **M** Heatmap summarizing Spearman correlations between neutrophils (CD66⁺CD32a⁺ neutrophils (total, immature, and mature) and either CX3CR1^high or CX3CR1^low macrophages. Spearman correlation coefficients (R) are represented by the color scale (red = positive correlations; blue = negative correlations) and by the size of the circles, which is proportional to the absolute value of R. All spearman correlation were two-sided. Statistical significance is indicated by asterisks placed inside each circle (\**p* < 0.05; \*\**p* < 0.01; \**p* < 0.001). Source data are provided as a Source Data file.

macrophage activation states may be partially coordinated across mucosal and nodal compartments. CD4⁺ T-cell parameters displayed a different pattern: total CD4⁺ T cell frequencies, CD4⁺ T-cell activation (HLA-DR⁺), proliferation (Ki67), and PD-1 expression were strongly correlated between intestine and lymph nodes (Fig. 8M–P). These

results indicate that CD4⁺ T-cell immune activation and dysfunction are systemically coordinated across compartments.

Plasma cytokine profiling revealed that IL-1Ra and IP-10 were significantly reduced in PTCs compared to SIV⁺ animals (q = 0.014 and 0.0039) while MCP1, ITAC, and IFNα2 were lower in PTCs relative to

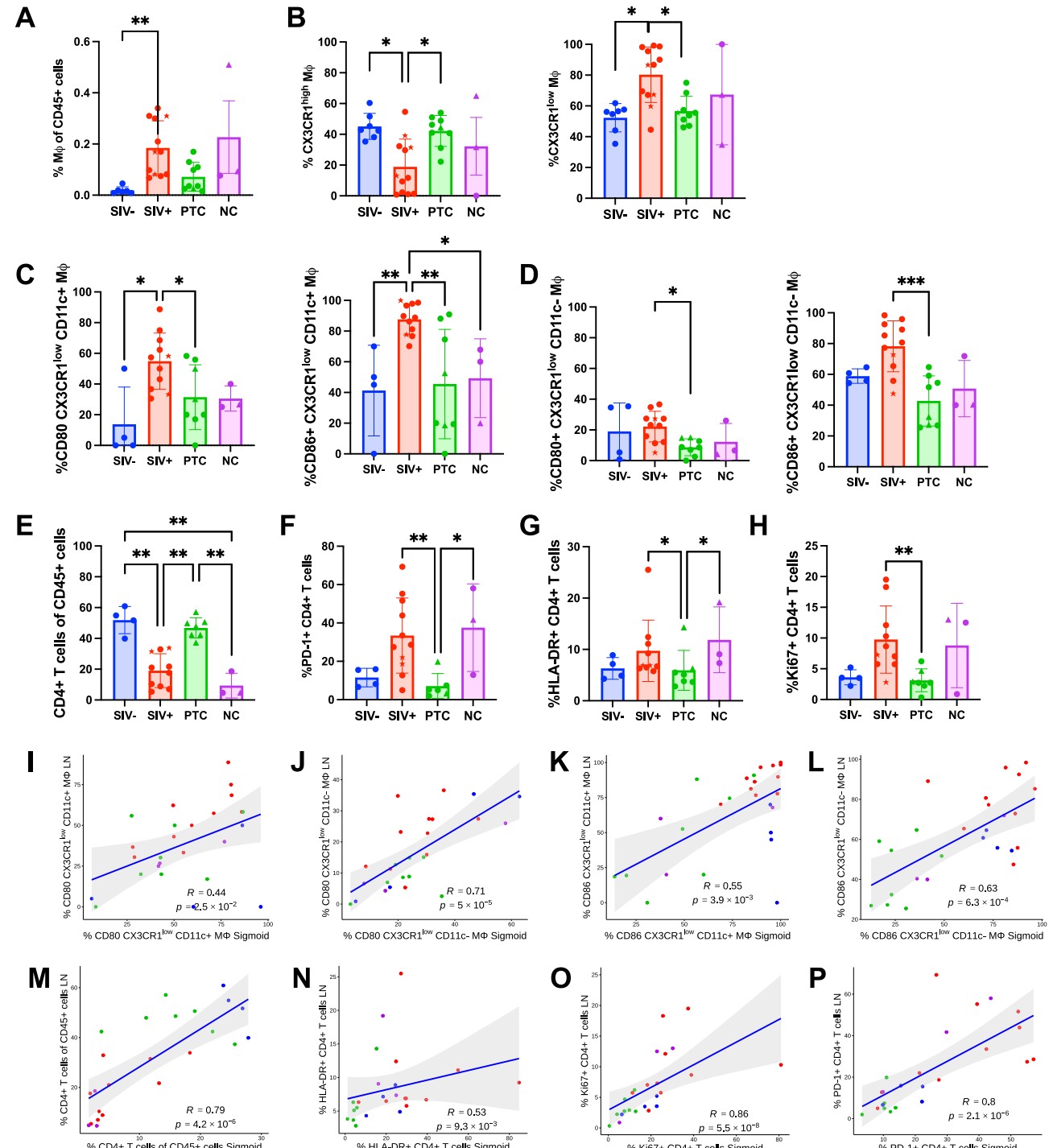

**Fig. 8 | Pro-inflammatory macrophage subsets and activated CD4+ T cell phenotypes predominate in the draining colon lymph nodes of viremic animals.** **A** Frequency of total macrophages among CD45+ lymph node cells. **B** Frequencies of CX3CR1high and CX3CR1low macrophages. **C** Frequencies of CD80+ and CD86+CX3CR1lowCD11c+ macrophages. **D**Frequencies of CD80+ and CD86+CX3CR1lowCD11c− macrophages. **E** Frequency of total CD4+ T cells among CD45+ lymph node cells. **F–H** Frequencies of PD-1+ (**F**), HLA-DR+ (**G**), and Ki-67+ (**H**) CD4+ T cells. All of these data were measured in the draining colon lymph nodes of SIV− (n = 7), SIV+ (n = 12), PTC (n = 9), and NC (n = 3) animals *Kruskal−Wallis with Benjamini−Krieger−Yekutieli (FDR) correction; q < 0.05 (*), q < 0.01 (**), q < 0.001*

(***). The results are presented as individual data points together with the mean ± standard deviation. **I–P** Scatter plots show Spearman correlations of cell frequencies measured in the sigmoid colon versus the corresponding colon-draining lymph nodes from the same animals: CX3CR1high and CX3CR1low macrophages (I-L) and multiple CD4+ T cell parameters (**M–P**): total CD4+, HLA-DR+, Ki-67+, and PD-1+, CD4+ T cells. Blue lines depict linear regression fits with 95% confidence intervals (gray shading). Confidence intervals are centered around the mean of the x value. Spearman correlation coefficients (**R**) and p-values are indicated in each panel. All spearman correlation were two-sided. Source data are provided as a Source Data file.

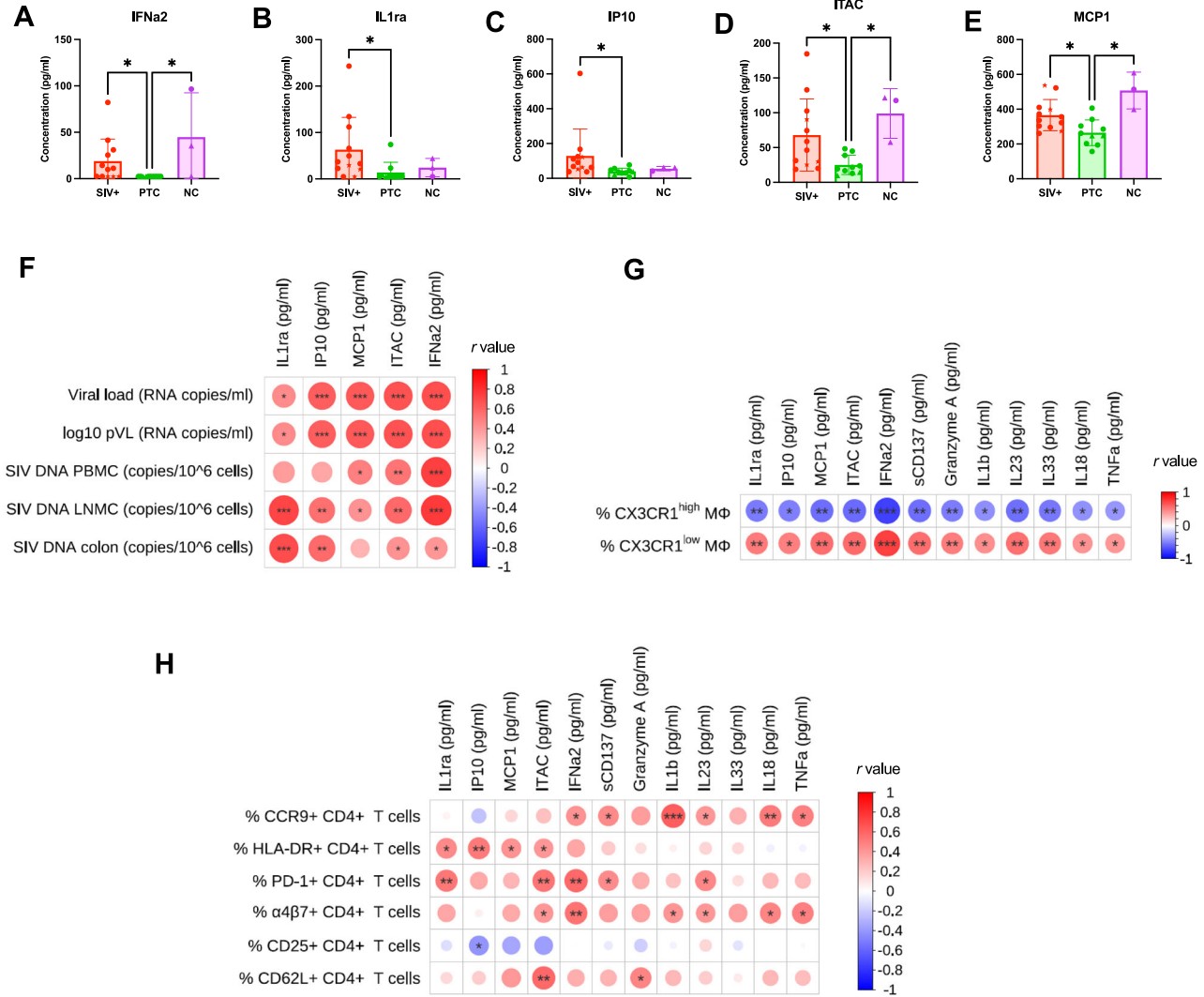

**Fig. 9 | Reduced levels of circulating inflammatory cytokines in post-treatment controllers correlate with lower viral burden, restrained macrophage activation, and limited CD4⁺ T cell activation. A–E** Plasma concentrations of IL-1RA (**A**), IP-10 (**B**), MCP-1 (**C**), ITAC (**D**), and IFN-α2 (**E**) across SIV⁺ (n = 12), PTC (n = 10), and NC (n = 3) animals. The results are presented as individual data points together with the mean ± standard deviation. *Kruskal–Wallis with Benjamini–Krieger–Yekutieli (FDR) correction; q < 0.05 (*), q < 0.01 (**), q < 0.001 (***).* **F** Heatmap of Spearman correlations between these five cytokines and plasma viral load as well as SIV DNA in PBMCS, LNMCs, and sigmoid colon. **G** Heatmap of Spearman correlations between a broader panel of 13 plasma cytokines (IL-1RA, IP-10, MCP-1, ITAC, IFN-α2,

sCD137, granzyme A, IL-1β, IL-23, IL-33, IL-18, and TNF-α) and the frequency of CX3CR1^high or CX3CR1^low macrophages. **H** Heatmap of Spearman correlations between the same cytokines and CD4⁺ T cell subsets expressing CCR9, HLA-DR, PD-1, α4β7, CD25, and CD62L in the sigmoid colon. Spearman correlation coefficients (R) are represented by the color scale (red = positive correlations; blue = negative correlations) and by the size of the circles, which is proportional to the absolute value of R. Statistical significance is indicated by asterisks placed inside each circle (*p < 0.05; **p < 0.01; *p < 0.001). All spearman correlation were two-sided. Source data are provided as a Source Data file.

both SIV⁺ and NC groups (q < 0.01 for all) (Fig. 9A–E). These cytokines positively correlated with pVL and SIV DNA in PBMCS, lymph nodes, and colonic tissues (Fig. 9F), as well as with CX3CR1^low Mφ (Fig. 9G). Additional pro-inflammatory and immunostimulatory mediators, including sCD137, granzyme A, IL-1β, IL-23, IL-33, IL-18, and TNF-α, followed similar trends with CX3CR1^low Mφ abundance (Fig. 9G). These cytokines also correlated with key markers of CD4⁺ T cell activation and mucosal homing (PD-1, HLA-DR, α4β7, and CCR9) (Fig. 9H), highlighting a coordinated activation network linking macrophage polarization to T cell dysfunction. Notably, this network was attenuated in PTCs.

## Discussion

Our study identifies the maintenance of intestinal CX3CR1⁺ Mφ homeostasis as a defining immunological signature linked to post-

treatment control in SIV-infected cynomolgus macaques. By leveraging a well-characterized NHP model with integrated analysis of mucosal, lymphoid, and cytokine compartments, we demonstrate that animals achieving durable viral remission after ART interruption maintain a balanced mucosal myeloid niche, characterized by the predominance of CX3CR1^high Mφ and a limited accumulation of activated CX3CR1^low subsets. This Mφ equilibrium was strongly associated with the preservation of CD4⁺ T cell homeostasis, attenuated neutrophil activation, and reduced systemic inflammation, together delineating an immune environment conducive to post-ART viral control. These findings position CX3CR1^high Mφ as central mediators of mucosal immune quiescence and potential therapeutic targets in HIV cure strategies.

Consistent with prior observations in HIV/SIV models[12,15,48], chronic infection in our study led to pronounced depletion of mucosal

CD4$^+$ T cells and accumulation of immune-activated phenotypes. Even in the setting of post-treatment control, CD4$^+$ T cell frequencies in the gut were only partially restored, underscoring the long-lasting impact of early viral damage on mucosal immune architecture[14,49]. Chronic infection was further associated with skewing of CD4$^+$ memory subsets toward Tscm and reduced CM populations, alongside increased PD-1 expression, indicative of T cell exhaustion, consistent with previous findings and further expanded here[50–54].

Although intestinal macrophage polarization has been studied in the context of HIV infection[55,56], the specific contribution of CX3CR1$^+$ Mφ subsets has been largely overlooked in both HIV and SIV studies. However, CX3CR1-expressing Mφ have emerged as key regulators of immune homeostasis and mucosal responses in various intestinal inflammatory conditions. In healthy gut tissue, CX3CR1$^{high}$ Mφ sustain immune quiescence, while CX3CR1$^{low}$ subsets are associated with inflammatory responses[22,28,31,33,57–61].

A central finding of this study is the phenotypic and functional polarization of intestinal Mφ based on CX3CR1 expression. In SIV$^-$ and PTC animals, Mφ exhibited a homeostatic CX3CR1$^{high}$ phenotype associated with tissue integrity and immune regulation. In contrast, chronic SIV infection induced a profound shift toward CX3CR1$^{low}$ Mφ, characterized by increased CD14 and CD11c expression, upregulation of costimulatory and homing molecules (CD40, CD83, CD86, α4β7), and broad associations with viral burden, and mucosal and systemic immune activation, supporting the view that CX3CR1 polarization evolves in parallel with ongoing viral replication and inflammation.

Several mechanisms may underlie the divergent activation profiles of CX3CR1$^{high}$ and CX3CR1$^{low}$ Mφ observed in our study. In the intestine, Mφ phenotype is shaped by local differentiation cues and by the balance between long-lived tissue-resident versus newly recruited monocyte-derived cells[30,62]. CX3CR1$^{high}$ Mφ corresponds to mature resident populations conditioned by IL-10 and TGF-β to adopt a regulatory, inflammation-refractory state[22,30,63]. In contrast, inflammatory environments promote continuous monocyte recruitment and differentiation into CX3CR1$^{low}$CD14$^+$CD11c$^+$ activated Mφ, a process enhanced by GM-CSF and type I interferon signaling[32]. These recruited cells retain heightened responsiveness to TLR ligands and produce stronger pro-inflammatory outputs, consistent with the activation signatures observed in SIV+ and NC animals. NC animals fail to restore this regulatory CX3CR1$^{high}$ compartment despite prolonged ART, consistent with sustained inflammatory monocyte influx and impaired differentiation during chronic SIV infection[64–66]. NC animals mirrored the inflammatory polarization seen in untreated chronic infection, although a complete overlay was not observed. Although the low frequency of intestinal macrophages precluded sorting-based functional assays, the combined phenotypic and cytokine correlation patterns suggest that CX3CR1$^{high}$ Mφ preservation reflects a more regulated immune environment after ART.

Microbial-derived signals are key modulators of intestinal CX3CR1$^+$ mononuclear phagocytes. CX3CR1$^+$ cells maintain epithelial integrity and coordinate mucosal immune responses through direct interactions with commensal microbes[67,68]. Commensals promote regulatory, tissue-protective programs, whereas dysbiosis can skew CX3CR1$^+$ cells toward pro-inflammatory states and impair barrier repair[27,69]. Through these pathways, CX3CR1$^+$ phagocytes influence Th17, Treg, and ILC3 homeostasis[25,27], positioning them as key intermediaries between microbial ecology and mucosal immunity. These observations raise the possibility that microbial cues contribute to the divergent macrophage polarization profiles observed here. Defining how microbiota-driven pathways shape CX3CR1$^+$ macrophage function during ART and after interruption represents an important avenue for future work.

These Mφ subsets were not only phenotypically distinct but also associated with divergent immune profiles. CX3CR1$^{high}$ Mφ displayed positive associations with Treg and Th17 frequencies, both key mediators of mucosal tolerance and barrier repair. In contrast, CX3CR1$^{low}$ were linked to increased CD4$^+$ T cell activation, proliferation, and exhaustion, and showed a trend toward Th1 expansion and enhanced CD4$^+$ T cell recruitment. Taken together, these associations delineate a mucosal immune environment in which inflammatory macrophage accumulation co-occurs with broader perturbation of CD4$^+$ T-cell homeostasis.

These observations align with prior reports linking intestinal Mφ polarization to mucosal immune regulation in inflammatory conditions and extend this concept to the context of SIV infection. In murine models, mature IL-10−producing CX3CR1$^{high}$ Mφ have been shown to promote immune tolerance and barrier integrity by supporting Treg and Th17 cell homeostasis[26,33]. Our findings corroborate this model and further delineate the phenotypic and functional divergence of intestinal Mφ subsets in SIV infection. By linking Mφ phenotype to CD4$^+$ T cell dynamics and immune activation, our study advances the understanding of how distinct macrophage niches may contribute to immune dysregulation and potentially influence viral persistence in chronic infection. These results complement existing models of mucosal immune dysfunction in HIV/SIV, highlighting a possible active role for Mφ subsets in shaping the adaptive immune landscape of the gut. Principal component analysis of colonic immune populations supported these findings, revealing distinct clustering of PTC animals from viremic SIV+ and NC animals, with CX3CR1$^+$ Mφ abundance and CD4$^+$ T cell subsets as major contributors to this separation, reinforcing their role as key immunological features associated with post-treatment control.

Although NC animals showed a statistically higher proportion of Th17 cells compared with chronically viremic SIV$^+$ animals, this difference was driven by an outlier subject and did not reflect a broader restoration of mucosal immunity. Th17 frequencies did not differ significantly among SIV$^-$, SIV$^+$, and PTC animals, indicating that Th17 preservation is not a defining feature of post-treatment control in this cohort. Given that pathogenic SIV infection is classically characterized by profound Th17 depletion in rhesus macaques[70–72], the levels observed in NC animals are unlikely to represent a biologically meaningful recovery. Instead, the inflammatory macrophage and CD4$^+$ T-cell profiles in NCs suggest persistent mucosal dysfunction despite marginal differences in Th17 frequencies. In contrast, regulatory T cells showed a markedly different pattern. PTC animals maintained mucosal Treg frequencies comparable to uninfected controls, whereas both SIV$^+$ and NC animals exhibited significant depletion. The impact of HIV/SIV infection on mucosal Tregs is known to be complex and compartment-specific[73], with studies reporting depletion in the intestinal *lamina propria* of SIV-infected pigtailed and cynomolgus macaques[74,75], and increased proliferation in the colonic mucosa of chronically infected rhesus macaques[76]. This variability underscores the influence of species, tissue compartment, infection stage, and immune environment. Our findings are consistent with observations in African green monkeys, where experimental Treg depletion during acute SIV infection led to sustained T cell activation but did not trigger mucosal inflammation or disease progression[77], supporting the notion that control of inflammation, rather than T cell activation alone, is the key determinant of non-pathogenic outcomes. Together, these data suggest that mucosal Treg preservation may contribute to the immune equilibrium observed in post-treatment controllers by limiting local inflammation and maintaining tissue integrity.

A moderate inverse correlation between CX3CR1$^{high}$ Mφ and gut-resident Tscm frequencies was also observed; however, the biological significance of this finding remains uncertain. Most studies identifying Tscm cells as long-lived viral reservoirs have focused on circulating Tscm[78], and the contribution of gut-resident Tscm to SIV persistence is not well defined. In the absence of direct virological measurements in sorted Tscm subsets, these cross-sectional associations cannot be interpreted in terms of reservoir size or post-treatment outcome.

Future longitudinal studies with tissue-resident reservoir quantification will be required to determine whether macrophage polarization influences the maintenance or decay of Tscm populations in the gut.

Our data reveal that neutrophil activation, evidenced by increased frequencies of CD11b[high] and CD32a[+] neutrophils[47,79], particularly within immature subsets, was prominent in viremic animals and associated with CX3CR1[low] Mφ abundance. This coordinated expansion suggests a myeloid inflammatory axis in which macrophage–neutrophil co-activation may amplify mucosal immune activation in the context of ongoing viral replication. Given prior reports implicating neutrophils in IFN-γ–mediated inflammation and tissue injury during HIV/SIV[41,79,80], this axis may constitute a key barrier to immune re-equilibration and mucosal healing following ART interruption.

The increased frequency of CD11b[high] neutrophils in NCs, paralleled by the reduction of CD11b[low] subsets, is consistent with inflammation-driven remodeling of neutrophil subsets described in HIV/SIV infection. CD11b upregulation is a well-established marker of neutrophil priming and heightened responsiveness, triggered by sustained type I/II interferon signaling, microbial translocation, and chronic mucosal inflammation. These conditions favor the accumulation of activated or primed neutrophils at the expense of less activated subsets[42,79,81–83]. In NC animals, the combination of delayed ART initiation and repeated inflammatory insults associated with viral rebound likely amplifies these pathways, resulting in a more pronounced neutrophil activation profile than in chronically viremic SIV+ animals. Together, these mechanisms provide a coherent explanation for the elevated CD11b[high] signatures in NCs, supporting the interpretation that persistent inflammatory pressures, rather than ART exposure alone, shape neutrophil activation patterns in this group.

This polarization signature extended beyond the gut, shaping the lymphoid and systemic immune landscape. In SIV+ and NC animals, coordinated expansion of CX3CR1[low] Mφ and activated CD4+ T cells occurred in both colon and draining lymph nodes, paralleled by upregulation of costimulatory molecules and exhaustion markers (PD-1, HLA-DR, Ki-67). In contrast, PTC animals exhibited lymphoid phenotypes resembling uninfected controls. Although the direction of macrophage skewing was shared across compartments, paired analyses demonstrated that the magnitude of CX3CR1[high/low] polarization was not quantitatively coordinated between intestine and lymph nodes, indicating that macrophage polarization is primarily shaped by local mucosal cues. In contrast, CD4+ T-cell activation and several activation-associated CX3CR1[low] Mφ subsets (CD80+/CD83+/CD86+) showed strong mucosal–lymph node correlations, underscoring a broadly coordinated systemic immune activation in viremic animals. This was echoed in plasma cytokine profiles: inflammatory mediators such as IL-1Ra, IP-10, MCP-1, ITAC, and IFNα2 were significantly elevated in viremic animals and correlated with CX3CR1[low] Mφ abundance, CD4+ T cell activation, and viral load. Together, these observations support a model in which intestinal macrophage polarization reflects a tissue-intrinsic immune milieu that intersects with systemic inflammatory pathways to shape post-ART outcomes.

We acknowledge that this study is limited by relatively small group sizes, particularly among late-treated NCs and in the subgroup of animals analyzed for neutrophil phenotypes. These limitations may reduce the statistical power to detect subtle immunological differences and warrant cautious interpretation of subgroup-specific findings. Nonetheless, the consistency of trends observed across both innate and adaptive immune compartments supports the biological relevance of the associations described, highlighting the need for validation in larger, longitudinal cohorts and mechanistic investigations.

Additionally, we relied on CX3CR1 expression as a surrogate marker of Mφ function, based on previously reported associations with regulatory or inflammatory roles[22,28,59,60]. While we did not directly assess cytokine production or transcriptional profiles within these subsets, the consistent upregulation of classical activation markers in CX3CR1[low] Mφ supports a pro-inflammatory interpretation. Nonetheless, future studies incorporating direct functional assays will be essential to clarify the inflammatory potential of these cells and distinguish between immune-activated and regulatory states within the gut microenvironment. In addition, although we observed a persistent skewing toward CX3CR1[low] Mφs in NCs post-ATI, the directionality of this association remains unsolved, and CX3CR1[low] Mφs may represent either contributors to, or biomarkers of, chronic immune activation.

A key limitation of this study is that tissue analyses were performed exclusively at necropsy, several months after ATI, precluding longitudinal assessment of Mφs dynamics during ART and immediately after treatment interruption. The cells present at necropsy may differ from those at ATI, and the current dataset does not capture the early immunological events that initially determined whether viral rebound occurred; therefore, causality cannot be inferred. Nonetheless, the long interval between ATI and necropsy enables the characterization of stable immune set-points. Given the uniform ART/ATI protocols and fixed virological outcomes, the mucosal and lymph node phenotypes detected here likely represent durable immune signatures of immune restoration rather than short-lived fluctuations.

Several observations strengthen the biological plausibility that preservation of CX3CR1[high] Mφ reflects a more favorable mucosal immune environment. A subset of untreated animals with low-level viremia ("natural controllers") exhibited the highest frequencies of CX3CR1[high] and lowest frequencies of CX3CR1[low] Mφs among SIV+ animals, suggesting that a homeostatic Mφs profile can emerge independently of ART. CX3CR1[high] Mφ abundance was also associated with regulatory CD4+ programs (Tregs, Th17), consistent with a less inflammatory mucosal environment. Moreover, early-infection data from our previous work show that the shift toward CX3CR1[low] Mφ begins within days of mucosal exposure[34], indicating that macrophage dysregulation arises early in SIV infection and may influence downstream immune trajectories rather than merely reflect chronic viral burden.

These innate features align with adaptive immune correlates reported in the extended pVISCONTI cohort[47], including enhanced stem-like, polyfunctional CD8+ T cells after early ART initiation, raising the possibility that CX3CR1[high] Mφ and antiviral CD8+ memory responses may act synergistically or in parallel to preserve mucosal integrity and limit viral reactivation after ART interruption. Definitive resolution of causality will require longitudinal mucosal sampling, particularly around ART cessation, and functional manipulation of macrophage subsets to determine whether CX3CR1[high] Mφ actively contribute to establishing post-treatment control or represent a stable immunological signature of individuals capable of maintaining viral remission.

It is worth noting that species-specific features of SIVmac251 infection in cynomolgus macaques are important for interpreting post-treatment outcomes. Although partial natural control has been described in this species, baseline frequencies remain low (~25% in this study and ~12% in the original pVISCONTI cohort[47]), and animals carrying MHC haplotypes associated with enhanced spontaneous control were excluded. Moreover, all animals were infected intravenously with a high viral dose to minimize early virological heterogeneity. A major strength of the cynomolgus macaque model is its strong alignment with human post-treatment control. The ART schedule used here was selected to mirror conditions inferred from the human VISCONTI cohort[6]. Notably, rebound kinetics in cynomolgus macaques closely resemble those reported in ART-interruption trials in people living with HIV[6,7,84,85], including similar time-to-rebound and post-rebound set-points. This parallelism reinforces the translational relevance of this model for studying mechanisms of durable remission. By contrast, rhesus macaques typically exhibit higher peak viremia, faster disease progression, and more extensive reservoir seeding, resulting in rapid

and uniform rebound unless ART is initiated extremely early[86–88]. Overall, these species-dependent characteristics highlight why cynomolgus macaques offer a relevant framework for interpreting post-ART outcomes in a way that aligns closely with human clinical data.

One recurring concern regarding the PTC phenotype is whether viral remission in the absence of ART might come at the cost of residual immune activation and long-term comorbidities. While prior studies have shown that PTCs tend to exhibit lower T cell activation and exhaustion compared to noncontrollers[84,89], heterogeneity exists, and residual low-level immune perturbation may persist. These findings underscore the importance of evaluating long-term immune health and comorbidity risks in PTC populations. In this context, our study provides additional insights by showing that PTC animals exhibit an immunological profile closely resembling that of uninfected controls across mucosal, lymphoid, and systemic compartments, suggesting that viral remission in this model does not come at the expense of persistent immune dysregulation.

Beyond virological control, these findings may have important clinical implications. Chronic immune activation and mucosal dysfunction are key drivers of non-AIDS comorbidities in people living with HIV despite suppressive ART. Our results raise the possibility that maintaining or restoring mucosal myeloid equilibrium could contribute not only to post-treatment viral remission but also to improved long-term health by limiting inflammation-driven complications. While further studies are warranted to directly assess clinical outcomes, our findings support the notion that preservation of intestinal CX3CR1+ Mφ homeostasis may serve as an indicator of durable immune health and reduced risk of inflammation-associated comorbidities in post-treatment remission. Targeting or monitoring this Mφ subset could therefore have important clinical implications for sustaining viral remission and mitigating chronic inflammation. These findings extend beyond the context of SIV infection to highlight a general principle: preservation of intestinal CX3CR1+ Mφ homeostasis appears essential for maintaining mucosal immune equilibrium, as previously shown in the context of SARS-CoV-2 infection[90].

In conclusion, our study provides novel evidence that intestinal Mφ polarization is a key correlate of post-treatment outcomes. These findings support a systems-level model in which the preservation of CX3CR1high Mφ reflects a state of mucosal immune homeostasis associated with limited T cell dysfunction and attenuated systemic inflammation, features that underpin durable post-treatment remission. Intestinal CX3CR1+ Mφ may therefore serve as biomarkers of mucosal immune regulation and represent candidate targets for therapeutic strategies aimed at sustaining viral remission.

## Methods

### Animals and ethical approvals

Cynomolgus macaques (CMs, *Macaca fascicularis*), originating from Mauritian AAALAC-certified breeding centers, were housed at the IDMIT infrastructure (CEA, Fontenay-aux-Roses) in AAALAC-accredited BSL-3 facilities (Animal facility authorization #D92-032-02, Préfecture des Hauts de Seine, France) and compliance with European Directive 2010/63/EU, French regulations, and the Standards for Human Care and Use of Laboratory Animals of the Office for Laboratory Animal Welfare under Assurance Numbers #A5826-01 and F20-00448. All work related to animals was conducted in compliance with institutional guidelines and protocols approved by the local ethics committee "Comité d'Ethique en Expérimentation Animale du Commissariat à l'Energie Atomique et aux Energies Alternatives" (CEtEA #44). The pVISCONTI study was approved and accredited under the statement A15 035 from the "Comité d'Ethique en Expérimentation Animale du CEA" and was registered and authorized under Number 2453-2015102713323361v3 by the French Ministry of Education and Research. The animals were housed in two separate rooms, each in social groups of 5/6 individuals, under controlled conditions of humidity, temperature, and a 12-hour light/dark cycle. They were fed commercial monkey chow and fruits once or twice daily, with water available ad libitum. Fifteen days before infection and until euthanasia, the SIV+ animals were housed individually and received enhanced food and structural enrichment. Environmental enrichment, including toys and novel food, was provided under the supervision of the CEA Animal Welfare Officer.

### Study design and sample utilization

Thirty-seven adult male CMs (median age = 5,2 years at inclusion, IQR = 4,3-6,5 years) were assigned to 4 groups: uninfected controls (Group 1/SIV-, $n = 12$), untreated SIV+ animals (Group 2/SIV+, $n = 12$) and 13 ART-treated animals with early (Group 3, $n = 9$) or late (Group 4, $n = 4$) initiation (Fig. 1A and Supplementary Table 1). ART was maintained for 2 years in both treated groups, followed by analytical treatment interruption (ATI). Animals were monitored for at least 6 months post-ATI, with a prescheduled study endpoint at 48 weeks after ATI. However, NC animals reached euthanasia criteria due to disease progression before the planned endpoint. Six out of 12 animals in Group 2, as well as animals from Groups 3 and 4, represent a subset of animals from the pVISCONTI study[47]. All animals tested negative for antibody responses to SIV, simian retrovirus type D (SRV), and simian T-cell lymphotropic virus (STLV), at the study's outset. CMs were intravenously exposed to 1000 animal infectious doses 50 (1000 AID50) of SIVmac251 isolate (kindly provided by Dr A.M. Aubertin, Université Louis Pasteur, Strasbourg, France)[91]. These experimental conditions ensured uniformly high plasma viremia during primary infection and precluded spontaneous SIV control[92]. Notably, animals bearing the M6 haplotype, previously associated with natural SIV control[93–95], were excluded from the study.

CMs initiated daily antiretroviral therapy approximately 4 weeks post-infection (Group 3) or 24 weeks post-infection (Group 4). This therapy consisted of emtricitabine (FTC, 40 mg/kg, Gilead), dolutegravir (DTG, 2.5 mg/kg, ViiV Healthcare), and the tenofovir prodrug tenofovir-disoproxil-fumarate (TDF, 5.1 mg/kg, Gilead) co-formulated for once-daily subcutaneous injection. Animals received daily ART for 24 months, then underwent analytic treatment interruption (ATI). Subsequently, animals were monitored for viral rebound (defined as the first viral load exceeding 400 SIV RNA copies after ART interruption) and/or post-treatment SIV control, as previously described[47]. Animals were sacrificed at the indicated time points by intravenous injection of 180 mg/kg of sodium pentobarbital (Doléthal, Laboratoire Vetoquinol).

### Sample collection and processing

Blood samples were collected in BD Vacutainer Plus Plastic K3EDTA tubes (BD Biosciences) for plasma viral load quantification before viral exposure from sedated animals following 5 mg/kg intra-muscular injection of Zoletil 100 (Virbac, Carros, France), during follow-up, and at necropsy. Plasma was isolated by centrifugation for 10 min at $480 \times g$ and cryopreserved at −80 °C.

Peripheral blood mononuclear cells (PBMCS) were isolated from Vacutainer CPT Mononuclear Cell Preparation Tubes with Sodium Heparin according to the manufacturer's instructions (BD Biosciences), and red blood cells were lysed in ammonium-chloride-potassium (ACK) buffer (0.15 M NH4Cl, 16 mM KHCO3, 0.1 mM EDTA, pH 7.4). The sigmoid colon and draining colon lymph nodes were collected at necropsy.

Lymph node single cells were obtained by mechanical dissociation, passing through a 70 µm nylon filter using gentle pressure applied with a sterile syringe plunger. *Lamina propria* mononuclear cells (LPMCs) were isolated from fresh intestinal tissues immediately after necropsy, as previously described[96]. Sigmoid colons were cut into small pieces and incubated for 20 min at 37 °C in HBSS medium without Ca++/Mg++ (Fisher Scientific, Illkirch, France) supplemented

with 5 mM EDTA and 1 mM DTT (Sigma-Aldrich, St Quantin Fallavier, France) to eliminate mucus and epithelial cells. After washing in PBS, the tissue was incubated for 1 h at 37 °C with agitation in HBSS medium with $Ca^{++}/Mg^{++}$ (Fisher Scientific, Illkirch, France) containing collagenase type VIII (0.25 mg/ml, Sigma Aldrich, St Quantin Fallavier, France) and DNase (5 U/ml, Roche, Mannheim, Germany). Undigested pieces were submitted to a second digestion for 30 min. Cell suspensions from both lymph nodes and colon were filtered through 70-µm sterile nylon cell strainers (BD Biosciences), washed with complete medium (RPMI supplemented with 10% FCS, 100 U/ml penicillin/streptomycin, 1% glutamine, 1% NEAA, 1% Na-pyruvate, 1% HEPES buffer [1 M]; all from Fisher Scientific, Illkirch, France), stained on the day of isolation for phenotypic characterization and stored at 4 °C overnight before flow cytometric acquisition the following morning. For functional assays, freshly isolated cells were stimulated overnight with PMA/Ionomycine (PMA at 62 ng/ml and Ionomycine at 720 ng/ml; all from Fisher Scientific, Illkirch, France) in the presence of Brefeldin A (10 µg/ml from Sigma Aldrich, St Quantin Fallavier, France) and stained the next day. No cryopreservation was applied at any step. Overall, cell recovery and viability were comparable across groups. The total number of isolated cells fell within a similar range for all conditions (mean ± SD: SIV- $85 \pm 26 \times 10^6$, SIV + $66 \pm 30 \times 10^6$, PTC $114 \pm 56 \times 10^6$, and NC $103 \pm 52 \times 10^6$), and viability remained consistently high, varying between approximately 67% and 80% (mean ± SD: $SIV^- 80 \pm 5\%$, $SIV^+ 72 \pm 5\%$, PTC $71 \pm 9\%$, NC $67 \pm 4\%$).

### Flow cytometry and cell staining
Nine multiparameter panels were developed. Four panels were designed to target leukocyte subpopulations and T-cell activation (P1 & 2), Treg cells (P3), and cytokine profile (P4) (Supplementary Table 2). Four panels were tailored for myeloid cell identification (P1) and activation-co-stimulation-trafficking status (P2, 3, 4) (Supplementary Table 3). One panel was developed for neutrophil analysis (Supplementary Table 4).

All staining was performed after saturation of Fc receptors using healthy 10% macaque serum (in-house production) for 1 h at 4 °C. Amine-reactive dye Live/dead Fixable Blue (Life Technologies) was used to assess cell viability and to exclude dead cells from the analysis. Four million cells were labeled from each sample with the antibody cocktail for 30 min at 4 °C, washed in PBS/10% FCS, and fixed in Cell-FIX™ (BD Biosciences), as previously described[96]. For Ki-67 and Foxp3 intracellular staining, cells were permeabilized using the Foxp3/Transcription Factor Staining Buffer Set (Invitrogen) and stained in 1x PermWash (eBioscience) containing the corresponding antibody for 1 h at 4 °C. T cell responses were also characterized by measuring the frequency of CD4 + T cells expressing IFN-γ, IL-17A, and IL-22. One million cells were cultured in complete medium and stimulated with PMA (62 ng/ml, Fisher Scientific) and Ionomycin (720 ng/ml, Fisher Scientific). Brefeldin A (Sigma-Aldrich) was added to each well at a final concentration of 10 µg/ml and the plate was incubated at 37 °C, 5% CO2 for 18 h. Next, cells were washed, stained with a viability dye, stained with anti-CD3, -CD4, and -CD8 antibodies, and then fixed and permeabilized (Cytofix, Cytoperm, BD). Cells were stained with anti-IFN-γ, -IL-17A, and -IL-22 antibodies. Corresponding isotype controls were used at the same concentrations as the reference antibody. Cells were acquired using a Fortessa X20 flow cytometer (BD Biosciences) and DIVA software. Batch-related variability was minimized by maintaining strict consistency in instrument setup and reagents. Daily QC using CS&T beads confirmed stable PMT target values across acquisition days, and each cohort was acquired using the same antibody lots whenever possible. When a new antibody lot was introduced, titration was repeated to ensure comparable staining performance. In addition, qualitative drifts in the major immune populations (e.g., CD3+ T cells, CD4+ T cells, CD14+ monocytes) were not observed across runs during manual gating inspection. Therefore, no additional batch-correction procedures were required. All results were analyzed using FlowJo 9.8.3 (Tristar, USA) software within the singlet viable fraction.

### Quantification of plasma viral load, SIV-DNA, and cell-associated SIV-RNA
Blood viral RNA was obtained from 100 µL cell-free plasma using the Nucleospin 96 RNA kit (Macherey Nagel GmbH&Co KG, Düren, Germany), according to the manufacturer's instructions. Retro-transcription and cDNA amplification and quantification were performed in duplicate by RT-qPCR using the Superscript III Platinum one-step quantitative RT-PCR system (Invitrogen, Carlsbad, USA). RT-PCR was performed as previously described[47]. The quantification limit (QL) was estimated to be 111 copies/ml and the detection limit (DL) was 12.3 copies/ml.

Total DNA and cell-associated RNA were extracted from PBMCS using an AllPrep DNA/RNA Mini Kit (Qiagen). For tissue samples, to control for differences in viral distribution within a single organ, two or three tissue samples were mechanically disrupted separately with a MagNA Lyser (GmbH, Roche Diagnostics, Brussels, Belgium). Nucleic acids were extracted separately, and the lysate was divided into two parts for DNA (DNA Mini Kit, Qiagen) and RNA extraction (RNeasy Plus Mini Kit, Qiagen). Cell-associated RNA was treated with DNase I during extraction (Qiagen). Viral measurements were then performed on each extract. Total SIV-DNA and SIV cell-associated RNA were quantified by ultrasensitive real-time PCR and by one-step real-time PCR targeting the gag region, respectively, as previously described[47,97]. The DNA and RNA thresholds varied according to the number of cells and the quantity of total RNA available and were calculated for each assay.

### Cytokines quantification
Plasma cytokines were measured in 25 µL serum using the Milliplex® Non-Human Primate Cytokine/Chemokine/Growth Factor Panel A Magnetic Bead Panel - Premixed 37-plex (Merck Millipore). Immunoassays were performed according to the manufacturer's instructions. Data were acquired using a Bio-Plex 200 instrument and analyzed using Bio-Plex Manager Software, version 6.2 (Bio-Rad).

### Data visualization and statistical analysis
All data visualization and statistical analyses were carried out using Prism v10.1.2 (GraphPad Software, La Jolla, USA) or R version 4.3.3 (http://www.R-project. org) with the corrplot and heatmap3 packages for heatmaps, the ggpubr package for correlation plots with confidence intervals, and the FactorMineR and factoextra packages for PCA. The results are presented as individual data points together with the mean ± standard deviation. For comparisons between two groups, a nonparametric Mann-Whitney U test was used for unpaired data, and a paired nonparametric Wilcoxon signed-rank test was used for paired data. p-values ≤ 0.05 were considered statistically significant. Differences across more than two groups were evaluated using the Kruskal-Wallis test, and p-values were corrected for multiple comparisons using the Benjamini, Krieger, and Yekutieli FDR approach; adjusted values are reported as q-values. Correlations were assessed using Spearman's rank analysis. Statistical significance was defined as $q \leq 0.05$ for two-tailed tests, with significance levels indicated as $*q < 0.05$, $**q < 0.01$, $***q < 0.001$, $****q < 0.0001$.

### Reporting summary
Further information on research design is available in the Nature Portfolio Reporting Summary linked to this article.

## Data availability
The data that support the findings of this study are presented in the main figures and supplemental material of this article. The raw data for graphs are available in the Source Data file. Source data are provided with this paper. Further information and requests for resources and

reagents should be directed to the lead contact. Source data are provided with this paper.

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

## Acknowledgements

We thank Caroline Passaes for sample preparation; Animalliance and animal care workers, in particular Sebastien Langlois, Benoit Delache, Claire-Maelle Fovet, Maxime Pottier, Jean-Marie Robert, Quentin Sconosciuti, Nina Dhooge, Emma Burban, Pauline Le Calvez, Raphaël Ho Tsong Fang and Quentin Pascal for NHP interventions; Jêrome Van Wassenhove, Wesley Gross, and Anne-Sophie Gallouet for flow cytometry assistance; Julie Morin, Kyllian Lheureux, Maxence Galpin-Lebreau, Mathis Lafosse, Laurine Moenne-Loccoz, Loic Pintore, Suzie Hecquet; Laura Junges, Dylan Wagner, Audrey Chatenet for PCR and immunological analysis; Mylinda Barendji, Julien Dinh, and Elodie Guyon for the management of NHP biological resources; Frédéric Ducancel, Alicia Pouget, Sylvie Legendre and Yann Gorin for their help with the logistics and safety management; Isabelle Mangeot and Salomé Piault for their help with resources management; Brice Targat and Karl-Stephan Baczowski for their contribution to data management, and Aurélien Marc for assistance with statistical analysis. FTC, DTG, and TDF were obtained from Gilead and ViiV Healthcare through the "IAS Toward an HIV Cure" common Material Transfer Agreement. The pVISCONTI study was funded by MSDAvenir through a research grant to the ANRS-RHIVIERA consortium, and the ANRS | Emerging infectious diseases French agency (ANRS-MIE). This work was funded by the ANRS-MIE, decision n° 20456, recipient MC. KB was supported by a Sidaction grant "Aides aux équipes". This work was also supported by the "Programme Investissements d'Avenir" (PIA) managed by the ANR under references ANR-11-INBS-0008 and ANR-10-EQPX-02-01, funding IDMIT infrastructure. The funders had no role in the design of the study, data collection, interpretation, or the decision to submit the work for publication.

## Author contributions

Study conception and design: MC, SH. pVISCONTI program design: ASC and RLG. Acquisition of the data: SH, KB, ML, LB, JL, FR, AM. Analysis and interpretation of the data: SH, KB, DD, ND, NB, VAF, and MC. Study management: DD and ND. Preparation of figures/tables: SH and MC. Study supervision: MC. Funding acquisition: MC, ASC, and RLG. Draft of the manuscript: MC with assistance from SH. All authors corrected and approved the final version of the manuscript.

## Competing interests

A.S.C. has received speaker fees from M.S.D., ViiV Healthcare, Gilead, and Janssen. V.A.F. has received grants (to her institution) from ViiV Healthcare and honoraria and travel grants from ViiV Healthcare and Gilead Sciences for participation in educational programs and conferences. All other authors declare that they have no competing interests.
