## [Peer Review file · Nature Communications]

Maintenance of intestinal CX3CR1⁺ macrophage homeostasis defines post-treatment control in SIV-infected macaques

Corresponding Author: Dr Mariangela Cavarelli

Version 0:

Reviewer comments:

Reviewer #1

(Remarks to the Author)

This manuscript by Hua et al., titled "Maintenance of intestinal CX3CR1⁺ macrophage homeostasis defines post-treatment control in SIV-infected macaques" presents a comprehensive analysis of mucosal macrophage polarization, T-cell dysregulation, and systemic inflammation in cynomolgus macaques intravenously infected with SIVmac251. The animals were divided into four groups: uninfected (SIV⁻), untreated infected (SIV⁺), ART-treated post-treatment controllers (PTC), and ART-treated non-controllers (NC). The study proposes that preservation of intestinal CX3CR1^{high} macrophages correlates with viral control following ART interruption. Overall, the work provides valuable insight into the role of intestinal macrophage polarization in post-treatment viral remission. However, interpretation of the findings is constrained by species-specific viral dynamics, the limited number of NC animals, and the absence of mechanistic validation. Several key issues require clarification or additional data to strengthen the study's scientific rigor and overall impact.

Although the authors used cynomolgus macaques to reduce MHC-related variability, this species is known to exhibit partial natural control of SIVmac251. Despite the exclusion of M6 haplotypes and confirmation of baseline seronegativity, the viral load kinetics (Fig. 1B-D) indicate relatively modest set-points. Moreover, ART suppression in all treated animals was apparently complete, but the extremely low rebound frequency after interruption (the post-ART NC:PTC ratio 1:8 in the early-treated group, Fig. 1C; especially 2:2 in the late-treated group, Fig. 1D) raises concern that spontaneous or host-intrinsic control may confound the classification of post-treatment controllers (PTCs). The authors should clarify whether viral suppression and rebound dynamics in this cynomolgus macaque model differ from those typically observed in rhesus macaques infected with SIVmac251, even SHIV strain (PMID: 25042999, 30575753, 34919814) and discuss whether the inherent control capacity of cynomolgus macaques could artificially inflate the apparent frequency of PTCs.

Although the manuscript provides adequate comparative data between SIV⁺ and PTC animals, it offers limited direct comparison between PTC and NC groups, particularly in Figures 3-5. Given that NCs are essential to testing the study's central hypothesis, expanding or reanalyzing these comparisons (even with the small sample size) would help determine whether macrophage polarization truly associates with post-treatment rebound outcomes rather than infection status alone. Based on the current data, intestinal macrophage polarization appears to correlate primarily with viral loads, but not directly with post-treatment viral rebound or sustained control and therefore may not serve as a specific biomarker for the PTC phenotype.

In Figure 2 (lines 151-156), the analysis of macrophage phenotypes is limited to comparisons between SIV⁺ and SIV⁻ animals. The rationale for excluding the PTC and NC groups at this stage is unclear and disrupts the continuity of the overall analysis. These groups should be incorporated into the initial macrophage profiling, or at least the authors should provide a clear justification for their exclusion.

The study presents a convincing association between CX3CR1^{high} macrophages and viral control; however, the supporting evidence remains largely correlative (line 185). For example, Figure 4 demonstrates opposing patterns of CD40 expression between CX3CR1^{high} and CX3CR1^{low} macrophages. The authors should elaborate on potential underlying mechanisms, such as altered differentiation cues (e.g., GM-CSF/IL-10 signaling), changes in monocyte recruitment, or microbial-driven modulation, which could explain this differential polarization. Incorporating cytokine production data (e.g., IL-10, IL-6, TNF- α)

or transcriptional profiling from sorted macrophage subsets would greatly strengthen the functional relevance of these findings.

The authors correlate macrophage subsets with Th17 and Treg frequencies (Fig. 6, lines 229-230), but do not extend this analysis to central memory (T_{cm}) and stem-cell memory (T_{scm}) CD4⁺ T cells, even though these subsets are presented in Figure 5. Given that T_{scm} cells represent key viral reservoirs, it would be valuable to examine whether CX3CR1^{high} macrophages correlate with the preservation or exhaustion of T_{cm} and T_{scm} populations, thereby providing a mechanistic link between innate immune regulation and reservoir maintenance.

In lines 255–259, the authors describe differences in neutrophil phenotypes, particularly the reduced frequency of CD11^{low} neutrophils in NCs compared with SIV⁺ animals (Fig. 7G–I), yet provide no mechanistic explanation for this observation. The authors should discuss potential factors contributing to this shift, such as ART exposure, mucosal barrier disruption, or alterations in myelopoiesis. Additionally, it would be valuable to assess whether CX3CR1^{low} macrophage abundance predicts neutrophil activation independently of viral loads, ideally through partial correlation or regression analyses.

In lines 306-315, the manuscript touches on macrophage activation within colon-draining lymph nodes but does not clearly relate these findings to mucosal macrophage phenotypes. A paired analysis between intestinal and nodal compartments (if available) could better define whether CX3CR1 polarization is locally restricted or systemically mirrored.

Additionally, the manuscript provides insufficient detail on the procedures for lamina propria cell isolation and cryopreservation. Given the inherent fragility of gut macrophages and dendritic cell subsets, the authors should clarify: (1) whether all gut samples were processed fresh or after cryopreservation; (2) how cell recovery and viability compared across the SIV⁺, PTC, and NC groups; and (3) whether batch effects were evaluated and corrected during flow cytometric acquisition.

Minor Comments

1. Convert numeric notation (e.g., Fig. 1H “9.1e-09”) to 9.1×10^{-9} for readability.
2. Move gating strategy (currently Fig. S2) into main Figure 2 for transparency.
3. “NCs” and “no-PTCs” refer to the same subset?
4. Figure 5A’s label “CD4⁺ T of CD45⁺” should follow a hierarchical gate (CD45⁺ → CD3⁺ → CD4⁺).
5. Lines 124-126, 133-140, 191, 259-260...q-values should be p-values.
6. Figure 2 legend appears to be missing the label “C.”
7. Align panels and legends across Figures 5 and 7.
8. Indicate whether bar plots represent mean ± SEM or mean ± SD.

Reviewer #2

(Remarks to the Author)

The manuscript by Hua et al. provides a comprehensive examination of immune phenotypes in gut and lymphoid tissues of SIV-infected cynomolgus macaques. Their observation that post-treatment controllers (PTCs) maintain intestinal CX3CR1-high macrophages while non-controllers exhibit inflammatory CX3CR1-low phenotypes, with strong correlation to viral burden, immune activation, and T cell preservation, is noteworthy. These findings reinforce the notion that preventing or limiting chronic inflammation during immunodeficiency virus infections leads to improved outcomes and suggest that CX3CR1-high macrophages may serve as a biomarker for post-treatment control. However, several points require clarification, and limitations need to be addressed before publication.

The study has important methodological constraints. Critically, all tissue samples were collected at necropsy after an extended time off ART. Consequently, the cells observed at this time point may differ from those that initially established post-treatment control or led to uncontrolled virus replication. While the authors appropriately acknowledge the lack of functional validation, the study remains purely correlative. It cannot determine whether CX3CR1-high macrophages actively create an anti-inflammatory environment or simply reflect reduced inflammation due to early ART and/or effective antiviral responses. The Discussion would also benefit from better contextualization of these findings within the broader landscape of post-treatment control, including the previously reported adaptive immune responses from the pVISCNTI cohort. With revisions addressing these concerns, the manuscript will be suitable for publication.

1. Three SIV⁺ animals that spontaneously controlled viremia (macaques 4, 7, and 12) are included in the SIV⁺ group for all comparisons, but they are not analyzed as a distinct subgroup. It is acknowledged in the Discussion that these natural controllers exhibited the highest frequencies of CX3CR1-high and the lowest frequencies of CX3CR1-low macrophages among SIV⁺ animals, suggesting a homeostatic profile similar to that of PTCs. Therefore, their inclusion in the SIV⁺ group may mask or dilute the true differences between chronically viremic and other groups. Please either discuss the natural controllers throughout the Results section, noting where they resemble PTCs versus viremic animals. Alternatively, justify why the natural controllers are grouped with the viremic animals for the comparisons.
2. All analyses were performed at necropsy after animals had been off ART for an extended period. The cells at this single time point may differ from those when post-treatment control was (or was not) established immediately after ART cessation. The inherent limitation of using necropsy samples should be more explicitly acknowledged when interpreting the results.
3. While the correlations between CX3CR1-high macrophages and favorable outcomes are strong. However, it remains

unclear whether these macrophages actively create the anti-inflammatory environment or were preserved due to early ART and/or effective antiviral responses. The observational study cannot distinguish whether these macrophages actively drive post-treatment control through immune regulation or are simply preserved in animals with effective antiviral responses. The proposed model, in which CX3CR1-high macrophages promote Tregs and sustain mucosal barrier integrity is attractive and plausible. However, the current study cannot determine causality. Please expand the Discussion to explicitly acknowledge this limitation and discuss how CX3CR1-high macrophages could work synergistically with adaptive immune responses to promote post-treatment control.

4. It is unclear why only four SIV-negative animals were used for the comparisons in Figure 1E, G, and H when twelve were available for subsequent analysis.

5. Line 114 states that PTC9 is in the early ART group, while Fig. 1D shows that this macaque is in the late ART group. Please clarify.

6. Figure 5J: While there is a statistical difference in the percentage of Th17+ cells between the SIV+ and NC groups, please discuss whether is likely a biological difference.

7. Lines 289 and 292: References to the supplementary figures appear to be incorrect. Line 289 should likely read "Supplementary Figure 11C," and line 292 should read "Supplementary Figure 11B."

Version 1:

Reviewer comments:

Reviewer #1

(Remarks to the Author)

The study by Charre et al. provides a comprehensive analysis of post-treatment control of SIVmac251-infected cynomolgus macaques model, offering insights into the mechanisms underlying sustained virological control after antiretroviral therapy interruption. Their findings include significantly lower SIV-DNA and cell-associated unspliced RNA levels in blood and lymph nodes of PTCs compared to NCs more than 6 months post-ATI, with early ART resulting in a higher PTC rate. They highlight the critical role of CD8+ T cells in selectively eliminating cells with intact proviruses in PLNs, with lower pre-ATI intact provirus levels in PLNs predicting PTC status. Additionally, intact provirus levels in blood CD4+ T cells 7 days post-ATI correlated with PLN reservoir levels, suggesting a potential biomarker for viral rebound. Limited viral evolution in PTCs, with proviruses genetically closer to the inoculum, further distinguishes them from NCs. These findings align with human studies like VISCONTI and CHAMP, extending our understanding of PTC mechanisms relevant to HIV remission strategies. In the latest version of this study, most of the questions and issues addressed have been answered and solved. One minor question is, the resolution of all figures need to be improved. Overall, we think this study is fine to publish.

Reviewer #2

(Remarks to the Author)

The authors have comprehensively addressed the concerns that I raised in the initial review. The revised manuscript is a rigorous and comprehensive analysis of mucosal macrophages in the cynomolgus macaque model of post-treatment control. The data are clearly presented, and the conclusions are now appropriately framed within the study's constraints. The study enhances our understanding of how innate immune cells contribute to post-treatment control and will be of significant interest to the HIV cure research community.

REVIEWER COMMENTS

Reviewer #1 (Remarks to the Author):

This manuscript by Hua et al., titled “Maintenance of intestinal CX3CR1⁺ macrophage homeostasis defines post-treatment control in SIV-infected macaques” presents a comprehensive analysis of mucosal macrophage polarization, T-cell dysregulation, and systemic inflammation in cynomolgus macaques intravenously infected with SIVmac251. The animals were divided into four groups: uninfected (SIV⁻), untreated infected (SIV⁺), ART-treated post-treatment controllers (PTC), and ART-treated non-controllers (NC). The study proposes that preservation of intestinal CX3CR1^{high} macrophages correlates with viral control following ART interruption. Overall, the work provides valuable insight into the role of intestinal macrophage polarization in post-treatment viral remission. However, interpretation of the findings is constrained by species-specific viral dynamics, the limited number of NC animals, and the absence of mechanistic validation. Several key issues require clarification or additional data to strengthen the study’s scientific rigor and overall impact.

We thank the reviewer for their positive assessment of our work and constructive comments.

1) Although the authors used cynomolgus macaques to reduce MHC-related variability, this species is known to exhibit partial natural control of SIVmac251. Despite the exclusion of M6 haplotypes and confirmation of baseline seronegativity, the viral load kinetics (Fig. 1B-D) indicate relatively modest set-points. Moreover, ART suppression in all treated animals was apparently complete, but the extremely low rebound frequency after interruption (the post-ART NC:PTC ratio 1:8 in the early-treated group, Fig. 1C; especially 2:2 in the late-treated group, Fig. 1D) raises concern that spontaneous or host-intrinsic control may confound the classification of post-treatment controllers (PTCs). The authors should clarify whether viral suppression and rebound dynamics in this cynomolgus macaque model differ from those typically observed in rhesus macaques infected with SIVmac251, even SHIV strain (PMID: 25042999, 30575753, 34919814) and discuss whether the inherent control capacity of cynomolgus macaques could artificially inflate the apparent frequency of PTCs.

We agree that cynomolgus macaques (CMs) can exhibit partial natural control of SIVmac251, although this occurs infrequently and is strongly influenced by MHC background (notably the M6 haplotype) and by specific experimental conditions such as low-dose or intrarectal infection (*Passaes et al Cell Reports 2020*). In the full pVISCONTI study, only 2 of 17 untreated animals (12%) maintained plasma viral loads <400 copies/mL during chronic infection, whereas the vast majority (88%) displayed high-level, persistent viremia (*Passaes et al., Nat Commun 2024*). In the present study, 3 of 12 untreated animals spontaneously controlled viremia, a proportion fully consistent with the previously documented baseline rate of natural control in this species.

Importantly, several methodological choices were specifically implemented to reduce inter-animal variability and minimize the impact of host-intrinsic control:

(i) animals bearing the M6 MHC haplotype, strongly associated with natural SIV control, were excluded; (ii) all animals were infected intravenously with a high dose of SIVmac251, ensuring highly homogeneous peak and set-point viremia; (iii) early- and late-ART groups were matched for all remaining MHC haplotypes.

These design elements markedly constrain immunogenetic heterogeneity and support the interpretation that differences in post-treatment outcome primarily reflect ART timing, not spontaneous or host-intrinsic control.

Consistent with the original pVISCONTI report, early ART initiation at week 4 induced high rates of post-treatment control (82% in the full cohort; 8/9 animals in the present subset), whereas late ART at week 24 resulted in markedly lower remission (18% in the full cohort; 2/4 animals here). These proportions are fully aligned with the biology of this model and are not indicative of unusually high spontaneous control.

We appreciate the reviewer's suggestion to contextualize rebound outcomes by comparison with rhesus macaque studies. We would like to clarify, however, that the experimental design of our study was intentionally aligned with conditions relevant to human post-treatment control. In particular, ART initiation at day 28 post-infection and the prolonged ART duration (2 years) were selected to mirror parameters inferred from the human VISCONTI cohort, in which early ART initiation favored the emergence of post-treatment controllers (*Saez Ciri3n et al., PlosPath 2013*). This design choice was deliberate: we aimed to model biological conditions under which post-treatment remission is observed in humans, rather than to recapitulate rebound dynamics classically described in rhesus macaques. Comparison with rhesus studies remains informative, yet must be interpreted in the context of well-established species differences in SIVmac251 pathogenesis, reservoir seeding, and responsiveness to ART. Rhesus macaques typically exhibit higher peak viremia, faster disease progression, and rapid, uniform rebound following ART interruption (PMID: 25042999, 30575753, 34919814), making this species poorly suited to study mechanisms of post-treatment control. In contrast, the magnitude of viral rebound in cynomolgus macaques closely resembles that documented in human ART-interruption studies (*Saez Ciri3n et al. PlosPath 2013*; Namazi G et al. *JID 2018*; *Ethemad B et al. PNAS 2023*). Importantly, viral set points among NCs in the original pVISCONTI cohort aligned closely with those reported in a recent meta-analysis of human ATI trials (*Gunst et al. Nature Comm 2025*). Thus, the CM model provides a biologically faithful window into human post-treatment outcomes, rather than an artificially permissive system.

Finally, the value of the cynomolgus model is underscored by the fact that key immunological signatures originally described in pVISCONTI macaques, particularly the expansion of stem-like

CD8⁺ T cells to counteract viral rebound following interruption of early ART, have been recently confirmed in human trials of treatment interruption (*Peluso et al., Nature 2025; Kiani et al., Nature 2025*). This concordance strongly supports the translational relevance of the CM model and reinforces that the mechanisms underlying PTCs after early ART in this model are a robust biological phenomenon, informative of the factors associated with viral control in humans.

We have clarified these points in the revised Discussion (lines 558-574).

2) Although the manuscript provides adequate comparative data between SIV⁺ and PTC animals, it offers limited direct comparison between PTC and NC groups, particularly in Figures 3-5. Given that NCs are essential to testing the study's central hypothesis, expanding or reanalyzing these comparisons (even with the small sample size) would help determine whether macrophage polarization truly associates with post-treatment rebound outcomes rather than infection status alone. Based on the current data, intestinal macrophage polarization appears to correlate primarily with viral loads, but not directly with post-treatment viral rebound or sustained control and therefore may not serve as a specific biomarker for the PTC phenotype.

We acknowledge the importance of direct comparison between PTC and NC animals. The intrinsically small size of the NC group (n=3) reflects the biology of the pVISCANTI model, as discussed in the previous comment, in which early ART initiation consistently generates a high frequency of post-treatment controllers. We had access to 3 NC animals for this study; therefore, increasing the NC numbers is not feasible.

Direct comparisons between PTC and NC animals were already included in Figures 3–5 through the four-group Kruskal–Wallis analyses, followed by appropriate post hoc comparisons between all groups, with correction for multiple testing. These analyses demonstrated consistent differences across all macrophage parameters, as well as CD4⁺ T-cell subsets, neutrophils, and cytokine levels between PTC and NC animals, even after correction.

To directly address the reviewer's request, we conducted an additional pairwise comparison between PTC and NC animals using a non-parametric Mann–Whitney test. We focused on the subsets that showed significant differences in the original Kruskal–Wallis analyses shown in Figures 3-5. All differences were confirmed using the Mann–Whitney test (**Figure R1**), except for total CX3CR1⁺ macrophages. Although this latter comparison did not reach statistical significance (exact $p = 0.0769$), the effect size remained considerable: PTCs displayed markedly higher frequencies of CX3CR1^{high} macrophages (median 75.45%) compared with NCs (median 57.60%). Despite the limited statistical power inherent to the small sample size, both the direction and magnitude of the difference were entirely consistent with the significant trends observed in the four-group Kruskal–Wallis analysis. Notably, frequencies of Th17 cells and HLA-DR⁺ CD4⁺ T cells, which were not significantly different between PTC and NC in the Kruskal–Wallis analysis, did reach significance in the Mann–Whitney comparison (**Figure R1 F and H**, respectively).

Given the overall consistency of Kruskal-Wallis and Mann–Whitney analyses, and given the constraints of sample size, we consider it more appropriate to present these pairwise results in the

rebuttal rather than incorporate them into the main manuscript, where the Kruskal–Wallis test already provides the most statistically robust assessment.

Figure R1. Pairwise comparison of immune cell subsets between PTC and NC. Frequencies of intestinal immune cell populations were compared between PTC (green) and NC (magenta) animals using a Mann–Whitney test. Each panel represents an individual subset previously identified as significantly different in the four-group Kruskal–Wallis analysis or selected for targeted pairwise evaluation. Bars show median \pm SD.

To address the reviewer’s concluding remark, we agree that viral load is a major driver of mucosal immune perturbation in chronic SIV infection. However, several observations from our dataset and from published work indicate that CX3CR1 polarization is not solely determined by contemporaneous viremia. First, NC animals remained fully aviremic during ART yet failed to restore CX3CR1^{high} macrophages, whereas spontaneous controllers, who are aviremic without ART, retained the most homeostatic macrophage profiles. Second, early-infection data from our previous work (Cavarelli et al., *iScience* 2022) demonstrate that the shift toward CX3CR1^{low} macrophages emerges within days of mucosal exposure, well before peak viremia or ART initiation, supporting the notion that macrophage dysregulation reflects early immune events. We now emphasize these points in the Discussion (lines 540-548). In this context, our use of the term “biomarker” in the manuscript refers specifically to CX3CR1^{high} macrophages as an indicator of preserved mucosal immune regulation, rather than as a marker uniquely predictive of the PTC phenotype.

3) In Figure 2 (lines 151-156), the analysis of macrophage phenotypes is limited to comparisons between SIV⁺ and SIV⁻ animals. The rationale for excluding the PTC and NC groups at this stage is unclear and disrupts the continuity of the overall analysis. These groups should be incorporated into the initial macrophage profiling, or at least the authors should provide a clear justification for their exclusion.

We understand the reviewer's confusion and appreciate the opportunity to clarify the rationale behind the structure of this section. The first paragraph of the Results defines the four cohorts included in the study, and the second examines the myeloid compartment across these groups. From this point onward, the manuscript focuses specifically on the role of intestinal macrophages, which represent the core of our study.

Before dissecting the macrophage differences observed following ART interruption, we considered it essential to first define how SIV infection alone alters intestinal macrophage homeostasis. Figure 2 was therefore intentionally designed to address this specific, introductory question. The corresponding section ("SIV infection disrupts intestinal macrophage homeostasis and skews CX3CR1 expression") establishes the foundational macrophage alterations induced by chronic SIV infection. This includes the detailed phenotyping of CX3CR1^{high} and CX3CR1^{low} macrophages and the characterization of infection-driven shifts in marker expression, an analysis that, to our knowledge, has not been previously reported in the context of HIV or SIV infection and thus represents an original observation central to the manuscript.

For this reason, the initial profiling in Figure 2 is restricted to the SIV⁺ vs SIV⁻ comparison, allowing us to define the baseline impact of infection on intestinal macrophage populations before introducing the additional layers of ART and analytical treatment interruption. Including PTC and NC animals at this stage would obscure infection-driven effects and reduce the clarity of a key observation that is original to this study. In the subsequent figures, all four groups (SIV⁻, SIV⁺, PTC, NC) are fully integrated to examine how ART timing and post-treatment outcome modulate these infection-driven alterations, thereby ensuring continuity in the overarching narrative.

We have now clarified this rationale in the revised manuscript (lines 144-147) to avoid any perceived discontinuity in the narrative.

That said, if the reviewer considers it necessary, we are willing to move the current Figure 2 to the Supplementary Information. We would, however, prefer to retain it in the main display, as these infection-driven alterations form the conceptual foundation for interpreting the ART- and ATI-related findings, and presenting them upfront maintains a clear and coherent flow throughout the manuscript.

4) The study presents a convincing association between CX3CR1^{high} macrophages and viral control; however, the supporting evidence remains largely correlative (line 185). For example, Figure 4 demonstrates opposing patterns of CD40 expression between CX3CR1^{high} and CX3CR1^{low} macrophages. The authors should elaborate on potential underlying mechanisms, such as altered differentiation cues (e.g., GM-CSF/IL-10 signaling), changes in monocyte recruitment, or microbial-driven modulation, which could explain this differential polarization.

Incorporating cytokine production data (e.g., IL-10, IL-6, TNF- α) or transcriptional profiling from sorted macrophage subsets would greatly strengthen the functional relevance of these findings.

We thank the reviewer for this insightful suggestion. We agree that the opposing activation profiles of CX3CR1^{high} and CX3CR1^{low} macrophages likely reflect distinct differentiation cues and tissue signals. Although a mechanistic dissection was beyond the scope of the present study, several pathways described in the literature may plausibly contribute to the phenotype we observe. We have now expanded the Discussion to elaborate on these potential mechanisms, supported by published evidence (lines 395-411).

We are particularly intrigued by the idea that microbial-driven modulation may represent a potential mechanism underlying macrophage polarization. Several studies have shown that dysbiosis, microbial translocation, and enhanced TLR stimulation can drive intestinal macrophages toward an inflammatory phenotype (Kim et al., *Immunity* 2018; Kim et al., *Gut Microbes* 2019), as reported in the Discussion (lines 412-421). Preliminary analysis, however, suggests that in our model, the magnitude of microbiota alterations is modest, consistent with previous studies in rhesus macaques showing limited dysbiosis during chronic SIV infection (originally demonstrated by the Brenchley group). For this reason, while microbial cues cannot be excluded, we do not consider them the primary driver of macrophage polarization in this dataset. A separate, dedicated analysis of the microbiota (including longitudinal samples across ART and ATI) is ongoing and will be reported in a subsequent manuscript.

We fully agree that functional assays (e.g., IL-10, IL-6, TNF- α production) or transcriptomic analyses of sorted CX3CR1^{high} / CX3CR1^{low} macrophages would provide valuable insight into the regulatory programs underlying their phenotype. Unfortunately, such analyses were not technically feasible in this study, mostly because of the low macrophage yield from intestinal tissue. In cynomolgus lamina propria, macrophages typically represent 1–2% of CD45⁺ cells. The absolute number of macrophages was insufficient to permit high-purity sorting or downstream RNA-seq without compromising other analyses (flow cytometry of 9 panels). We have added a short clarification in the Discussion acknowledging these technical constraints while outlining these functional investigations as an important future direction (lines 408-411; 521-526).

5) The authors correlate macrophage subsets with Th17 and Treg frequencies (Fig. 6, lines 229-230), but do not extend this analysis to central memory (Tcm) and stem-cell memory (Tscm) CD4⁺ T cells, even though these subsets are presented in Figure 5. Given that Tscm cells represent key viral reservoirs, it would be valuable to examine whether CX3CR1^{high} macrophages correlate with the preservation or exhaustion of Tcm and Tscm populations, thereby providing a mechanistic link between innate immune regulation and reservoir maintenance.

We thank the reviewer for this suggestion. In response, we performed the requested analyses examining whether CX3CR1^{high/low} macrophage frequencies correlate with central memory (CM) or stem-cell memory (Tscm) CD4⁺ T-cell subsets. No significant association was observed

with CM cells. A statistically significant negative correlation was detected with Tscm frequencies (**Figure R2**) and is shown in the revised Figure 6 of the manuscript as panels G, H and related Results section (lines 244-249); however, the biological interpretation of this finding is uncertain, as the role of gut-resident Tscm in SIV persistence is not well defined, and we did not perform direct reservoir quantification in this subset. Most studies identifying Tscm cells as major HIV/SIV reservoirs have focused on circulating Tscm populations, whereas the contribution of intestinal Tscm to reservoir maintenance remains largely unexplored. In the absence of functional or virological measurements in sorted Tscm subsets, cross-sectional correlations at the necropsy time point cannot be meaningfully linked to reservoir size.

We have added a sentence to the Discussion (lines 468-475) acknowledging that dedicated longitudinal studies with direct reservoir measurements will be required to determine whether macrophage polarization influences the composition or reservoir potential of long-lived tissue-resident memory CD4⁺ T-cell subsets.

Figure R2. Spearman correlations between total CX3CR1^{high} or CX3CR1^{low} macrophages and Tscm CD4⁺ T cell frequencies in the sigmoid colon.

6) In lines 255–259, the authors describe differences in neutrophil phenotypes, particularly the reduced frequency of CD11b^{low} neutrophils in NCs compared with SIV⁺ animals (Fig. 7G–I), yet provide no mechanistic explanation for this observation. The authors should discuss potential factors contributing to this shift, such as ART exposure, mucosal barrier disruption, or alterations in myelopoiesis. Additionally, it would be valuable to assess whether CX3CR1^{low} macrophage abundance predicts neutrophil activation independently of viral loads, ideally through partial correlation or regression analyses.

We agree that the reduced frequency of CD11b^{low} neutrophils in NC animals compared with chronically infected SIV⁺ animals warrants further clarification, and we have expanded the Discussion to address potential mechanisms underlying this shift (lines 484-495).

To address the second point raised by the reviewer, we performed partial correlation analyses adjusting for plasma viral load. After adjustment, the macrophage–neutrophil correlations were no longer significant, indicating that neutrophil activation primarily reflects ongoing viral replication rather than being independently predicted by macrophage abundance. Accordingly, we have interpreted these associations as reflecting coordinated inflammatory responses driven by viral replication, rather than direct mechanistic interactions, and we have been careful not to imply causality in the Discussion.

7) In lines 306-315, the manuscript touches on macrophage activation within colon-draining lymph nodes but does not clearly relate these findings to mucosal macrophage phenotypes. A paired analysis between intestinal and nodal compartments (if available) could better define whether CX3CR1 polarization is locally restricted or systemically mirrored.

To address this point, we performed a paired analysis comparing intestinal and colon-draining lymph-node macrophage phenotypes. No significant correlation was observed for the total, CX3CR1^{high} or CX3CR1^{low} macrophage subsets, indicating that the magnitude of CX3CR1 polarization is shaped primarily by local mucosal cues rather than quantitatively mirrored across compartments. In contrast, several activation-associated subsets of CX3CR1^{low} macrophages (CD80⁺, CD83⁺, CD86⁺; CD11c⁺ or CD11c⁻ expressing those markers) showed significant positive correlations between the intestine and lymph nodes (Spearman $r = 0.42$ – 0.70 , $p = 0.0347$ – <0.0001), suggesting that inflammatory activation programs can be coordinated across tissues even when the underlying CX3CR1 polarization axis remains tissue-restricted.

We performed the same paired analysis for CD4⁺ T-cell parameters. Total CD4⁺ T-cells ($r = 0.79$, $p = < 0.0001$), as well as CD4⁺ T-cell activation (HLA-DR⁺; $r = 0.53$, $p = 0.0093$), proliferation (Ki67⁺; $r = 0.86$, $p < 0.0001$), and PD-1 expression ($r = 0.81$, $p < 0.0001$) showed strong and significant correlations between intestine and lymph nodes. These findings indicate that CD4⁺ T-cell immune activation and dysfunction represent systemically coordinated immune processes, in contrast to the locally regulated CX3CR1 polarization program.

We have incorporated these results into the revised manuscript (lines 337-348 and **Figure 8I-P and Supplemental Figure 12**) and expanded the Discussion to clarify how mucosal and nodal immune programs relate to one another (lines 501-507). Importantly, we refined the wording of the original Discussion section to avoid implying fully coordinated macrophage polarization across compartments, while preserving the conclusion that systemic immune activation is a hallmark of SIV⁺ and NC animals and is mitigated in PTCs.

8) Additionally, the manuscript provides insufficient detail on the procedures for lamina propria cell isolation and cryopreservation. Given the inherent fragility of gut macrophages and dendritic cell subsets, the authors should clarify: (1) whether all gut samples were processed fresh or after cryopreservation; (2) how cell recovery and viability compared across the SIV⁺, PTC, and NC groups; and (3) whether batch effects were evaluated and corrected during flow cytometric acquisition.

We have clarified these aspects in the revised manuscript. The detailed protocol for LPMC purification was previously published by our group (*Benmeziane et al, STAR Protocols 2022*), and we included this reference in the revised Materials and Methods to give readers complete information.

1) All lamina propria samples were processed from fresh tissues. Immediately after LPMC isolation, cells were either (i) stained on the same day and kept at 4 °C overnight before acquisition, or (ii) stimulated overnight for functional assays and stained the next morning. In all cases, samples were never cryopreserved at any stage. This workflow was specifically optimized to preserve fragile intestinal macrophage and dendritic-cell subsets, whose viability and marker stability are markedly compromised by freeze–thaw cycles. This is now clarified in the Material and Methods (lines 664–666; 677–679; 682).

2) We systematically quantified both cell recovery and viability for all animals included in the study. Cell recovery from intestinal tissues was comparable across groups (mean ± SD, ×10⁶ cells): SIV⁻: 85 ± 26; SIV⁺: 66 ± 30; PTC: 114 ± 56; NC: 103 ± 52. Similarly, cell viability assessed by trypan blue exclusion was consistently high and did not differ substantially between groups exposed to the virus, but was higher in uninfected macaques (mean ± SD, % viable cells): SIV⁻: 80 ± 5; SIV⁺: 72 ± 5; PTC: 71 ± 9; NC: 67 ± 4. As requested, this information has been integrated into the revised Material and Methods section (lines 682–686).

These data indicate that tissue processing efficiency and cell integrity were equivalent across all groups, excluding cell recovery or viability as potential confounders in the interpretation of our immunological findings. For transparency, we provide the corresponding plot in the response file (**Figure R3**), but we prefer not to include this figure in the main manuscript, as these metrics serve as quality controls rather than primary endpoints.

Figure R3. Number of cells recovered from the sigmoid colon following tissue digestion (A) and cell viability determined by Tripin blue exclusion (B) among the four groups of animals.

3) We minimized batch-related variability by maintaining strict consistency in instrument setup and reagents. Daily QC using CS&T beads confirmed stable PMT target values across acquisition days, and each cohort was acquired using the same antibody lots whenever possible. When a new antibody lot was introduced, titration was repeated to ensure comparable staining performance. In addition, we did not observe qualitative drifts in the major immune populations (e.g., CD3⁺ T cells, CD4⁺ T cells, macrophages, etc) across runs during manual gating inspection. Therefore, no additional batch-correction procedures were required. That information has been integrated into the revised manuscript (lines 709-716).

Minor Comments

1.Convert numeric notation (e.g., Fig. 1H “9.1e-09”) to 9.1×10^{-9} for readability.

Numeric notation has been changed.

2.Move gating strategy (currently Fig. S2) into main Figure 2 for transparency.

The gating strategy figure has been moved to Figure 2 as requested.

3.“NCs” and “no-PTCs” refer to the same subset?

We confirm that “NCs” and “no-PTCs” would indeed refer to the same subset. The definition “no-PTCs” was used in *Passaes et al.* original study. In the present study, after re-checking the manuscript, we realized that the term “no-PTCs” is not used anywhere in the text or figures. Only “NCs” is used throughout to denote non-controllers.

4. Figure 5A's label "CD4⁺ T of CD45⁺" should follow a hierarchical gate (CD45⁺ → CD3⁺ → CD4⁺).

The label of Figure 1A has been changed as suggested. It reads now "% CD4⁺ T cells of CD45⁺ CD3⁺ cells"

5. Lines 124-126, 133-140, 191, 259-260... q-values should be p-values.

In the sections mentioned, we report *q-values* rather than *p-values* because all group comparisons were corrected for multiple testing using the Benjamini, Krieger, and Yekutieli false discovery rate (FDR) method. As per statistical best practices, when FDR adjustment is applied, the resulting values are q-values, not p-values, and we therefore indicate them explicitly. To avoid confusion, we have now clarified this point in the Methods (lines 748-755) and kept the notation "q-value" throughout the manuscript and figure legends when appropriate. p-values are instead displayed when two groups were compared, as described in the Methods.

Thanks to the reviewer's comment, we realized the statistical test was not always appropriately indicated in the figure legends. This has been corrected.

6. Figure 2 legend appears to be missing the label "C."

We thank the reviewer for noticing. This has been corrected.

7. Align panels and legends across Figures 5 and 7.

Panels have been aligned in Figures 5 and 7

8. Indicate whether bar plots represent mean ± SEM or mean ± SD.

All bar plots in the manuscript represent mean ± SD, as now indicated in the figure legends and clarified in the Methods section.

Reviewer #2 (Remarks to the Author):

The manuscript by Hua et al. provides a comprehensive examination of immune phenotypes in gut and lymphoid tissues of SIV-infected cynomolgus macaques. Their observation that post-treatment controllers (PTCs) maintain intestinal CX3CR1-high macrophages while non-controllers exhibit inflammatory CX3CR1-low phenotypes, with strong correlation to viral burden, immune activation, and T cell preservation, is noteworthy. These findings reinforce the notion that preventing or limiting chronic inflammation during immunodeficiency virus infections leads to improved outcomes and suggest that CX3CR1-high macrophages may serve as a biomarker for post-treatment control. However, several points require clarification, and

limitations need to be addressed before publication.

The study has important methodological constraints. Critically, all tissue samples were collected at necropsy after an extended time off ART. Consequently, the cells observed at this time point may differ from those that initially established post-treatment control or led to uncontrolled virus replication. While the authors appropriately acknowledge the lack of functional validation, the study remains purely correlative. It cannot determine whether CX3CR1-high macrophages actively create an anti-inflammatory environment or simply reflect reduced inflammation due to early ART and/or effective antiviral responses. The Discussion would also benefit from better contextualization of these findings within the broader landscape of post-treatment control, including the previously reported adaptive immune responses from the pVISCONTI cohort. With revisions addressing these concerns, the manuscript will be suitable for publication.

We thank the reviewer for their positive evaluation of our work and constructive comments.

1. Three SIV+ animals that spontaneously controlled viremia (macaques 4, 7, and 12) are included in the SIV+ group for all comparisons, but they are not analyzed as a distinct subgroup. It is acknowledged in the Discussion that these natural controllers exhibited the highest frequencies of CX3CR1-high and the lowest frequencies of CX3CR1-low macrophages among SIV+ animals, suggesting a homeostatic profile similar to that of PTCs. Therefore, their inclusion in the SIV+ group may mask or dilute the true differences between chronically viremic and other groups. Please either discuss the natural controllers throughout the Results section, noting where they resemble PTCs versus viremic animals. Alternatively, justify why the natural controllers are grouped with the viremic animals for the comparisons.

The three spontaneous controller animals were retained within the SIV+ group because the study design and predefined group structure were based on ART exposure and post-ATI outcome, rather than on spontaneous viral control. We now make this point explicit in the Results section (lines 106-108).

We agree, however, that the phenotype of these spontaneous controllers is biologically informative. For this reason, they were consistently labeled with a distinct symbol (a star) in all figures. Following the reviewer's suggestion, we now explicitly comment on their phenotype at the relevant points in the Results section (lines 185-188; 222-224).

As noted in the Discussion, spontaneous controllers displayed the highest frequencies of CX3CR1high macrophages and the lowest frequencies of CX3CR1low macrophages among SIV+ animals, a profile closer to PTCs and SIV- animals. While this phenotype diverges from the expected SIV+ pattern, these animals do not dilute the group-level differences: in macrophage analyses (e.g., Fig. 3), they simply cluster toward the homeostatic end of the spectrum without altering the statistical significance of the comparisons.

In other analyses (e.g., Figs. 4, 5, 7, and 8), spontaneous controllers distribute similarly to the rest of the SIV+ group, indicating that they do not systematically shift or mask group-level patterns.

To address the reviewer’s concern, we performed a sensitivity analysis excluding the three spontaneous controllers from the SIV+ group. The resulting q-values are presented in **Table R1**. Importantly, all main findings remained unchanged: the depletion of CX3CR1high macrophages and the enrichment of CX3CR1low inflammatory macrophages in SIV+ animals remained significant after excluding spontaneous controllers. No changes were observed when analyzing CD4+ T cell subsets. These results confirm that the observed disruptions in macrophage homeostasis are robust and driven by the chronically viremic phenotype, and are not diluted by the inclusion of spontaneous controllers.

We did not include this post hoc analysis in the manuscript, as it does not alter the predefined group structure or the study’s conclusions.

	With controllers	Without controllers	With controllers	Without controllers	With controllers	Without controllers	With controllers	Without controllers	With controllers	Without controllers	With controllers	Without controllers
	Macro CX3CR1high		Macro CX3CR1low		Macro CD14+ CX3CR1high		Macro CD14+ CX3CR1low		Macro CD14- CX3CR1high		Macro CD14- CX3CR1low	
SIV- vs. SIV+	<0.0001	<0.0001	<0.0001	<0.0001	<0.0001	<0.0001	<0.0001	<0.0001	<0.0001	<0.0001	<0.0001	<0.0001
SIV- vs. PTC	0.535	0.5001	0.3062	0.3065	0.1728	0.1538	0.1674	0.1483	0.1911	0.2545	0.1845	0.2456
SIV- vs. NC	0.0197	0.0174	0.0204	0.0179	0.0091	0.0086	0.009	0.0084	0.0261	0.0237	0.0291	0.0266
SIV+ vs. PTC	0.0002	0.0006	0.0003	0.0009	0.0009	0.0021	0.001	0.0021	0.0038	0.0022	0.0037	0.0021
SIV+ vs. NC	0.5832	0.6156	0.3062	0.3232	0.3689	0.3612	0.3689	0.3611	0.5751	0.5139	0.5351	0.4717
PTC vs. NC	0.0597	0.0582	0.0478	0.0466	0.0493	0.0523	0.0501	0.0532	0.1832	0.1396	0.2015	0.156
	CD4+ T cells in CD45+		CM CD4+ T cells		Tscm CD4+ T cells		Th1 CD4+ T cells		Th17 CD4+ T cells		Treg CD4+ T cells	
SIV- vs. SIV+	<0.0001	<0.0001	0.1044	0.081	0.0754	0.0846	0.0259	0.0416	0.0912	0.0564	0.0156	0.0128
SIV- vs. PTC	0.0068	0.0056	0.6392	0.6468	0.0808	0.0777	0.2692	0.2651	0.6361	0.6306	0.0532	0.0534
SIV- vs. NC	<0.0001	<0.0001	0.0248	0.028	0.1149	0.1557	0.622	0.6173	0.1042	0.1035	0.0074	0.0099
SIV+ vs. PTC	0.004	0.005	0.1044	0.081	0.0003	0.0006	0.1819	0.2155	0.1042	0.0763	0.0003	0.0004
SIV+ vs. NC	0.0592	0.0855	0.1044	0.185	0.6684	0.8312	0.0654	0.0724	0.0111	0.0068	0.0853	0.1344
PTC vs. NC	0.004	0.005	0.0248	0.028	0.0057	0.0088	0.2692	0.2651	0.0912	0.0763	0.0003	0.0005

Table R1. Sensitivity analysis of group comparisons with and without spontaneous controllers.

q-values (Kruskal–Wallis test with FDR correction) obtained with and without the three controller animals in the SIV+ group. The table summarizes all pairwise comparisons across major macrophage and CD4+ T-cell populations.

2. All analyses were performed at necropsy after animals had been off ART for an extended period. The cells at this single time point may differ from those when post-treatment control was (or was not) established immediately after ART cessation. The inherent limitation of using necropsy samples should be more explicitly acknowledged when interpreting the results.

We agree that analyses performed exclusively at the necropsy time point cannot fully capture the early immunological events occurring immediately after ART interruption, when post-treatment control or rebound is first determined. A statement acknowledging this limitation was already present in the Discussion; we have now expanded it further to clarify its implications for data interpretation (lines 530-538)

3. While the correlations between CX3CR1-high macrophages and favorable outcomes are strong. However, it remains unclear whether these macrophages actively create the anti-inflammatory environment or were preserved due to early ART and/or effective antiviral responses. The observational study cannot distinguish whether these macrophages actively drive post-treatment control through immune regulation or are simply preserved in animals with effective antiviral responses. The proposed model, in which CX3CR1-high macrophages

promote Tregs and sustain mucosal barrier integrity is attractive and plausible. However, the current study cannot determine causality. Please expand the Discussion to explicitly acknowledge this limitation and discuss how CX3CR1-high macrophages could work synergistically with adaptive immune responses to promote post-treatment control.

We thank the reviewer for raising this important conceptual point. In line with the previous comment, we fully agree that the present study, based on observational data obtained at necropsy, cannot determine whether CX3CR1^{high} macrophages actively promote post-treatment control or are preserved as a consequence of early ART and/or effective antiviral adaptive responses. As such, the causal direction cannot be resolved within the constraints of the current experimental design (lines 526-529; 532-538; 553-557). Following the reviewer's recommendation, we have now expanded the Discussion to explicitly restate this point and to place CX3CR1^{high} macrophage homeostasis within the broader context of innate–adaptive cooperation in post-treatment control (lines 549-557). Together, data from *Passaes et al.* and the present datasets support a model in which CX3CR1^{high} macrophages are part of a broader innate–adaptive axis that stabilizes post-treatment control.

4. It is unclear why only four SIV-negative animals were used for the comparisons in Figure 1E, G, and H when twelve were available for subsequent analysis.

We thank the reviewer for pointing out this inconsistency. We would like to clarify that in Figure 1E, twelve SIV- animals were indeed included in the analysis, but their values clustered tightly along the axis, rendering individual data points difficult to visualize in the figure. For panels 1G and 1H, only four SIV- animals were shown because these two panels presented *blood* CD4⁺ T-cell data, whereas panels I-J-K in Figure 1, as well as all other Figures, reported *intestinal* measurements. At the time of the initial submission, full blood CD4⁺ datasets were not available for all SIV- animals. We have now completed this analysis and incorporated the available blood CD4⁺ T-cell data from 11 SIV- macaques into the revised figures; data for one additional animal were not available. These updates have been implemented in the revised version of Figure 1G and H.

5. Line 114 states that PTC9 is in the early ART group, while Fig. 1D shows that this macaque is in the late ART group. Please clarify.

We apologize for the confusion. PTC9 is a late-treated animal, as indicated in Fig.1D. We have corrected the text in the Results (line 115).

6. Figure 5J: While there is a statistical difference in the percentage of Th17⁺ cells between the SIV⁺ and NC groups, please discuss whether is likely a biological difference.

Although the NC group displayed a statistically higher frequency of Th17 cells compared with chronically viremic SIV⁺ animals, the modest magnitude of this difference, mostly driven by one outlier animal, limits its biological impact. Data from PTC vs NC cohorts examining Th17 cells are scarce. Strongin et al. (J. Virol. 2020) reported a higher frequency of Th17 cells in PTCs

compared to NCs in rectal tissues from Rhesus macaques infected with SIVmac239. Extensive literature shows that pathogenic SIV infection of Rhesus macaques is characterized by Th17 depletion (Brenchley et al., 2008; Klatt et al., 2012; Cecchinato et al., 2008). In our study, NC animals retained a strongly inflammatory mucosal environment, with persistent CX3CR1^{low} macrophage polarization and heightened CD4⁺ T-cell activation, indicating that they did not achieve genuine mucosal immune restoration. The observed increase in Th17 cells in NCs likely reflects residual variability rather than a biologically meaningful recovery of Th17-mediated barrier function. We now clarify this interpretation in the manuscript (lines 444-452).

7. Lines 289 and 292: References to the supplementary figures appear to be incorrect. Line 289 should likely read “Supplementary Figure 11C,” and line 292 should read “Supplementary Figure 11B.”

We thank the reviewer for noticing it. The order of the panels in Supplementary Figure 11 has been changed to keep the right order in the text.